# Brain Gamma-Stimulation: Mechanisms and Optimization of Impact

**DOI:** 10.3390/biology14121722

**Published:** 2025-12-01

**Authors:** Konstantin V. Lushnikov, Dmitriy A. Serov, Maxim E. Astashev, Valeriy A. Kozlov, Alexander Melerzanov, Maria V. Vedunova

**Affiliations:** 1Institute of Biology and Biomedicine, Lobachevsky State University of Nizhny Novgorod, 23 Gagarin Avenue, 603022 Nizhny Novgorod, Russia; kolushn1@yandex.ru (K.V.L.); mvedunova@yandex.ru (M.V.V.); 2Prokhorov General Physics Institute of the Russian Academy of Sciences, Vavilov Str. 38, 119991 Moscow, Russia; dmitriy_serov_91@mail.ru (D.A.S.); astashev@yandex.ru (M.E.A.); 3Federal Research Center “Pushchino Scientific Center for Biological Research of the Russian Academy of Sciences”, Institute of Cell Biophysics of the Russian Academy of Sciences, 3 Institutskaya St., 142290 Pushchino, Russia; 4Department of Fundamental Sciences, Bauman Moscow State Technical University, 5 2nd Baumanskaya St., 105005 Moscow, Russia; 5Institute of General Pathology and Pathophysiology, 8 Baltiyskaya St., 125315 Moscow, Russia; m83071@gmail.com

**Keywords:** gamma stimulation, visual stimulation, neural network, Alzheimer’s disease, dementia

## Abstract

Changes in the brain’s electrical rhythm at a frequency of 30–60 Hz (γ-rhythm) play an important role in cognitive activity, learning, problem solving, and memory. Changes in γ-rhythm are associated with a wide range of nervous system diseases (Alzheimer’s disease, dementia, and others). The use of periodic stimuli (light, sound, and their combination) affects γ-rhythm and is considered a potential means of preventing and/or treating nervous system disorders. A large amount of data has now been accumulated on this issue, and the existing knowledge needs to be systematized. This paper analyses the effectiveness of periodic stimuli on the γ-rhythm depending on the frequency of stimulation, its nature (light, sound), additional properties (color, method of exposure), species affiliation, and age of the ‘test subjects’. This article provides an overview of the ways in which periodic stimulation changes the γ-rhythm and suggests ways to increase the effectiveness of γ-rhythm stimulation in humans.

## 1. Introduction

Non-invasive physical methods are becoming increasingly prevalent in a variety of sectors, including industry, agriculture, and medicine. These modern approaches are being incorporated into our lives and are gradually replacing traditional chemical methods [1,2,3]. In this instance, the development of personalized, non-invasive medicine occupies a significant position. As our species name, *Homo sapiens*, suggests, the primary factor contributing to an individual’s quality of life is mental well-being and the capacity to maintain high cognitive function throughout life [4,5]. In cases of severe age-related changes and neurodegenerative diseases, this issue goes beyond mere comfort to encompass the fulfillment of fundamental needs and the preservation of life itself. Cognitive decline associated with aging is becoming an increasingly serious public health problem. It is estimated that, by the year 2050, 152.8 million people worldwide will be living with dementia [6]. Although there are numerous treatment options for cognitive impairment and dementia, a comprehensive cure remains elusive [7]. Consequently, there is an urgent need to develop additional treatments that can help to alleviate the personal, social and economic burdens associated with the increasing number of dementia diagnoses [8]. Rhythmic oscillations in the brain can be detected using local field potential, electrocorticography, electroencephalography, and magnetoencephalography. These oscillations are generated by the activity of different neuronal populations (neuronal ensembles) in various brain regions. These oscillations occur at the following frequencies: delta (1–4 Hz), theta (4–8 Hz), alpha (8–12 Hz), beta (15–30 Hz), gamma (30–90 Hz), and high gamma (>50 Hz) [9,10]. They control (or are a product of) the synchronization of neural impulses at the micro level, and at the macro level they ensure communication between different parts of the cortex, coordinating the temporal and spatial connectivity of the brain [11,12]. The γ-rhythm ensures the integrative activity of the brain during sensory, cognitive, and executive processes in humans and animals [13,14,15]. It is well established that the γ-rhythm plays a role in the processing of information during synchronization in the 35–85 Hz range, a process that occurs between distant regions of the brain [16,17].

The use of periodic stimuli to trigger the γ-rhythm has been the subject of extensive research for a long time. The first studies of this phenomenon date back to the 1950s. The following diagram illustrates the current rate of global publications concerning γ-stimulation (Figure 1).

Overall publication activity saw active growth from 1980 to 2000. The number of published papers remained steady between 2010 and 2020, before declining gradually between 2020 and 2025. This observed trend may highlight several important points. Firstly, the empirical knowledge base on this topic may have reached saturation. Secondly, there is a lack of general quantitative patterns that could accurately predict and select the most promising areas. Despite the overall decline in publication activity, specific areas of γ-stimulation application continue to experience active growth. Notably, the number of publications focusing on the using of γ-stimulation to prevent age-related brain changes, or as a potential therapy for neurodegenerative diseases is increasing. Papers devoted to neurodegenerative diseases currently account for up to a third of the total pool of papers on γ-stimulation, indicating growth in applied research in this area.

There is sparse data in the global literature on the effectiveness of γ-stimulation. However, there are review articles devoted to systematizing the available data, identifying the relationship between effectiveness and stimulus characteristics, and analyzing side effects [18,19]. Furthermore, a significant body of work has been published on the molecular and cellular mechanisms of action, as well as on individual aspects that increase their effectiveness. However, no systematic studies have been conducted to describe clear, quantitative relationships between the effectiveness of γ-stimulation and characteristics such as the central frequency of the stimulus, the stimulus type, the wavelength for visual stimulation, the age of the subjects, and the species (when studying other animals).

This review aims to partially fill this gap. The aim of the study was to identify key parameters determining the effectiveness of γ-stimulation, and to assess the contribution of these parameters to the overall effectiveness of the intervention. Individual aspects of γ-stimulation’s effectiveness were also assessed. This review will also provide an overview of γ-stimulation methods, the molecular and cellular mechanisms of γ-stimulation, the relationship between the γ-rhythm and key brain systems, and its potential therapeutic effects. It will also propose avenues for optimizing γ-stimulation.

## 2. General Functions of γ-Rhythms, Mechanism of Regulations

Gamma oscillations, characterized by a fast rhythm, allow excitation in a neural network to temporarily escape subsequent inhibition. This increases the efficiency, precision and selectivity of communication between different areas of the brain, a phenomenon known as interareal coherence [20]. Studies have shown that interneurons expressing parvalbumin (PV+) and somatostatin (SST+) play a key role in maintaining gamma activity in the cerebral cortex and hippocampus. The frequency of gamma oscillations is primarily determined by GABA receptor-induced inhibitory postsynaptic currents (Figure 2) [21,22,23].

It is currently suggested that, in different areas of the brain, there are fundamental mechanisms of the neural network responsible for storing and processing information, as well as supporting attention, cognition, and memory [12,19,24,25,26]. These rhythms have been recorded and studied in the cerebral cortex, the hippocampus, the amygdala, the olfactory bulb, the striatum, and the brainstem. Their role in controlling visual attention, object perception, sensory processing, short-term memory, retaining relevant information in working memory, encoding words, making cross-modal semantic comparisons, movement, and emotion has been demonstrated [12,19]. Figure 2 shows the results of our own experimental studies. It is clear that the parietal and occipital regions respond most actively to stimulation. In our own studies (as yet unpublished), we also tested the brain’s response to visual stimulation in the range of 30 to 50 Hz and found that different EEG leads recorded responses to different frequencies (32, 35, 42, 47 Hz, etc.). The response is highly individual. Therefore, the patterns of response of different brain regions to gamma stimulation at different frequencies remain to be fully described and elucidated.

By contrast, disruption to gamma oscillations leads to abnormal neural activity and brain dysfunction [4]. The γ-oscillations are altered in various brain diseases, including Alzheimer’s disease (AD), Parkinson’s disease, stroke, schizophrenia, and autism spectrum disorders [27,28,29,30,31,32]. Studies suggest that gamma oscillations may serve as potential biomarkers of neural imbalance or interneuron dysfunction, reflecting the underlying pathophysiological mechanisms of critical neural functions in neuropsychiatric diseases such as Alzheimer’s and Parkinson’s diseases [12,19,33,34,35]. Restoring normal gamma activity could be a therapy to improve higher-order cognitive functions, sensorimotor integration, working memory, attention, perceptual binding and network synchronization. Considerable evidence suggests that γ-synchronization affects neural circuit function and behavior [34,36,37,38]. γ-synchronization, whether performed by non-invasive or invasive methods, has been shown to have a powerful neuroprotective effect in brain diseases [39,40,41].

Neural synchronization is defined as the process by which the neural activity of a subject aligns with the frequency of repetitive sensory rhythms. This phenomenon can be observed as an increase in the electroencephalogram (EEG) spectrum power at the excitation frequency, otherwise referred to as the steady-state response [42]. At present, a variety of stimulation methods are employed for the purpose of inducing the γ-rhythm. These include sensory stimulation (auditory, visual, and tactile), optogenetics, transcranial electrical or magnetic stimulation, and deep brain electrical stimulation [19,35].

For over three decades, electrophysiologists have been aware of the relationship between the severity of Alzheimer’s disease (AD) and changes in the amplitude of electroencephalogram (EEG) oscillations at a frequency of 40 Hz [43]. The interest in intrinsic, induced, and 40 Hz-evoked oscillatory neural potentials and their relationship with cognitive functions arose even earlier [16,44,45,46,47,48,49,50,51,52]. In the following decades, and until recently, a significant number of publications described the results of studies that confirmed the importance of oscillations with a frequency of 40 Hz.

Furthermore, visual stimulation at 40 Hz has been demonstrated to enhance reaction time in a target detection task [53]. Subsequent research revealed a preference for frequencies of 10, 20, 40, and 80 Hz on the part of neural oscillators when human synchronous oscillations were measured in response to stimulation at 1–100 Hz [54]. In 2009, significant data were obtained on the importance of rapidly excitable PV+ interneurons for the emergence and maintenance of γ-oscillations, primarily in the 40 Hz range [53,54]. Later, the role of neuronal plasticity of CP-AMPARs- and mGluRs-dependent calcium signaling of PV+ interneurons and their interaction with SST+ interneurons in maintaining γ-oscillations was clarified [55,56,57].

The utilization of neurostimulation at 40 Hz as a therapeutic modality for the remediation of spectral aberrations in Alzheimer’s disease (AD) has emerged as a promising avenue for research. In 2016, researchers began to propose the use of γ-training using sensory stimulation (GENUS system, Gamma ENtrainment Using Sensory Stimuli) in the form of visual stimulation with stroboscopic flickering at 40 Hz for the treatment of AD [24,58]. The GENUS system utilized white light-emitting diodes (LEDs) with a correlated color temperature of 4000 Kelvin (K), exhibiting flicker at a frequency of 40 hertz (Hz) and a duty cycle of 50 per cent, with the objective of providing visual stimulation. The temporal modulation is of a stroboscopic nature, meaning that the flickering occurs with a modulation depth of 100%. In a series of experiments involving validated mouse models of AD, in which the light characteristics remained constant, GENUS therapy demonstrated a favorable neuroprotective effect [19,24,25,26,30,58,59,60,61]. Its utilization has been demonstrated to curtail amyloid production and enhance its elimination from the hippocampus and prefrontal cortex. Moreover, it has been shown to augment recognition and visual–spatial memory subsequent to “therapy” comprising daily 1 h sessions for one or more weeks [37,62].

Synchronized light and sound stimulation at 40 Hz has been shown to effectively induce corresponding brain activity at the same frequency [12,61]. Overall, 40 Hz brain stimulation is associated with reduced neuroinflammation, enhanced synaptic transmission, and increased expression of genes associated with synaptic plasticity. These effects improve cognitive function by activating intracellular signaling pathways [63,64,65,66,67,68,69]. GENUS at 40 Hz has been shown to increase cytokine expression in microglia, normalize circadian rhythms and decrease the amount of amyloid plaques [58,63,70]. Furthermore, other studies have demonstrated that multisensory stimulation at 40 Hz enhances the glymphatic clearance rate of Aβ via the glial clearance pathway [70].

Research is currently underway to identify the most effective treatment protocols and determine the mechanisms of gamma stimulation. It is important to carefully consider the treatment parameters.

This review will not provide detailed information on the types of γ-stimulation and their effects on various diseases. These are described in great detail in a recent comprehensive review [19]. Here, our aim is to focus on the general quantitative dependence of gamma stimulation effectiveness on stimulus and response system characteristics, mechanisms of action, and issues of gamma stimulation optimization.

## 3. Literature Data Analysis Protocol

### 3.1. Criteria for Including Publications in the Analysis

The work represents a quantitative synthesis or meta-analytical review based on previously published data. A literature search was performed using the publicly available databases PubMed (https://pubmed.ncbi.nlm.nih.gov/), Google Scholar (https://scholar.google.com/), and the websites of major publishers: Nature (https://www.nature.com/), MDPI (https://www.mdpi.com/), and others. The search was performed using the keywords ‘gamma stimulation’, ‘gamma frequency stimulation’, or ‘50 Hz brain rhythm stimulation’. To refine the search, we used combinations with additional keywords, such as: ‘gamma stimulation’ + ‘disorders’ (Alzheimer’s disease, Parkinson’s disease, etc.), ‘gamma stimulation’ + ‘aging’, ‘gamma stimulation’ + ‘sleeping’, ‘gamma stimulation’ + ‘sound’ or ‘light’, ‘gamma stimulation’ + ‘human’ or ‘animals’, etc. When including articles in the analysis, preference was given to articles published in the last 10 years (2015–2025) and 20 years (2005–2025). The shares of these works were 62% and 89%, respectively. Works published earlier (before 2005) were included if they contained pioneering data, important results for understanding the mechanisms of gamma rhythm functioning, or results that were not published in later periods. Publications that included completely identical experimental conditions to those already added, or those with insufficiently well-described experimental conditions (duration, frequency, nature of the stimulus, and its characteristics) were excluded from the analysis.

### 3.2. Data Processing

This review used a quantitative analysis that included papers which clearly specified the following stimulus characteristics: central frequency; stimulus source; exposure time; participant age; and status (healthy participants, patients, or animal models, with detailed inclusion and exclusion criteria). Since the effects of gamma stimulation vary in nature, direction and amplitude, comparing them with each other or constructing quantitative diagrams is significantly complicated. Additionally, it is impossible to plot gene expression (a.u.), dendrite branch length (μm) and field potential (mV) on a single graph. To overcome these limitations, we employed a well-established approach [71]. All quantitative changes described in the papers were expressed as percentages relative to the control, and then the absolute value of this was taken. Since some effects could be in the single percent range while others could be in the hundreds, the resulting values were log-transformed (see Appendix A).

As described previously, to enable comparison of disparate effects, we first converted to dimensionless quantities and calculated the effects of the treatments. We calculated the effects using Formula (1):(1)$$Effect\;modulus={log}_{10}\left(\left|\left(\frac{A-\Delta A}{A}\right)\times 100\%\right|\right),$$
where *A* is the initial value of the measured characteristic (EEG potential, proportion of differentiated cells, mental performance scores, etc.), and *ΔA* is the value of the same parameter after γ-stimulation (in the same units as *A*). Expressing the values as percentages allows us to convert to dimensionless quantities and compare the effectiveness of γ-stimulation for different targets (see below, Section 4.5). Taking the absolute values allows us to adequately average the contributions of disparate effects. Further logarithmization allowed us to adequately assess the significance of statistical differences between groups, where effect sizes vary from units to hundreds of percent. The initial quantitative values of *A* and *ΔA* for the analysis were taken from the texts and/or graphs of the analyzed manuscripts.

The calculated data were processed using non-parametric statistics. We used a Kruskal–Wallis ANOVA with a post hoc Dunn’s test to evaluate the significance of statistical differences. If the differences in median values between treatment groups were greater than would be expected by chance (*p* < 0.05), we proceeded with a pairwise multiple comparison using Dunn’s test (all pairwise comparisons without control). *p*-values were provided for the results of the Dunn’s test. To assess the significance of the data in the case of constructing 3D maps, we calculated z-scores according to the Formula (2)(2)$$z_{i}=\left(x_{i}-µ\right)/\sigma$$
where *z_i_* is z-score of individual *i* value, *x* is individual *i* value, μ is mean, σ is standard deviation. Statistically significant differences are those in which *z_i_* ≥ 2.

## 4. Dependence of the Effectiveness of γ-Stimulation on the Characteristics of the Stimulus and Responding Systems

### 4.1. Dependence of the Effectiveness of γ-Stimulation on Species

In the first stage of our analysis, we analyzed published data on the general effectiveness of the γ-stimulation method in various animal species. The effects of γ-stimulation were most frequently assessed in humans (61% of the analyzed data) and mice (35%). Results obtained in rhesus macaques were described less frequently (4% of the analyzed data). Due to the high variability in the direction and amplitude of the effects, we opted for a logarithmic scale (log_10_), expressed in absolute terms. According analyzes γ-stimulation was generally more effective in humans than in mice and marmosets (see Figure 3). This may indicate the presence of interspecies peculiarities in the generation and control of the γ-rhythm. The cognitive abilities of humans and other animals differ significantly. These differences may also affect the effectiveness of γ-stimulation. Secondly, behavioral and cognitive tests are easier to implement in humans and survey methods can be used. Physiological measurement methods (skin microhemodynamics, etc.) are also much simpler to implement in humans as immobilization and/or anesthesia are not required. We believe that these two factors enable us to obtain more accurate results in humans than in other animals.

### 4.2. Dependence of the Efficiency of γ-Stimulation on the Type of Stimulus

As previously mentioned, the main methods of delivering gamma stimulation are light (presentation of a visual stimulus or exposure to a flickering light source), sound stimuli, or a combination of both. As the effectiveness of γ-stimulation differs between humans and animals, we analyzed all quantitative data separately for humans and mice, as these are the most accessible and popular methods. Magnetic stimulation can sometimes be used in humans. The results of a comparison of different γ-stimulation methods are shown in Figure 4 below.

In mice, no difference in the effectiveness of gamma stimulation with light, sound or a combination of the two. Stimulation with sound stimuli was generally more effective than magnetic stimulation (see Figure 4). Light stimulation tended to be more effective than sound stimulation without light.

Light stimulation comes in two forms: visual stimulation via the visual analyser or transcranial stimulation. The second method involves irradiating selected brain regions with infrared radiation, which can penetrate cranial tissue, or with visible light through a transparent window created in the skull (for mice only). A comparison of the effectiveness of light γ-stimulation by exposure method is shown in Figure 5 below.

No differences were found between transcranial and visual γ-stimulation in mice. This lack of difference may be due to the paucity of visual stimulation for rodents, which typically consists of only periodic blinks. In humans, visual stimulation is significantly more effective than transcranial stimulation (~10 times). Visual γ-stimulation is almost ~5 times more effective in humans than in mice. The differences may be due to methodological considerations. Firstly, the skull thicknesses of humans and mice differ, making transcranial stimulation difficult to achieve in humans. Secondly, as noted above, visual stimulation in mice is typically limited to simple periodic blinks. More complex (and likely more effective) stimuli may be used in humans.

Three types of LED are typically used: broadband white with peaks in the 500–600 nm range, short-wavelength UV-blue (360–450 nm) and red/IR (600–1000 nm and above). There is evidence that the effectiveness of γ-stimulation depends on the wavelength of the stimulating light. Our analysis confirmed this concept. Light with a central wavelength of 500–600 nm was most effective for humans compared to deep red (>600 nm) and IR light (see Figure 6). Interestingly, visual stimulation with conventional white light is less effective for mice than for humans. The reduced efficacy of blue light for gamma stimulation compared to white light is consistent with classical concepts [36,72]. However, traditionally, white and far-red colors are considered to be equally effective for γ stimulation. Our data indicate the greater efficacy of white light (sometimes containing near-red) compared to red. This phenomenon is easily explained by the higher sensitivity of the human eye to yellow-green light compared to red. Moreover, this sensitivity is expressed not only at the quantum level of the eye’s receptor pigments, but also in the sensitivity of the central nervous system and its responses to color stimulation [73,74,75].

To further illustrate the relationship between gamma stimulation effectiveness and light wavelength, we attempted to plot straight lines of approximation (see Figure 7). We detected a trend of decreasing gamma stimulation effectiveness with increasing wavelength. This trend was observed for both mice and humans. However, the R^2^ value was too small to consider the ‘wavelength-effectiveness’ relationship significant. Perhaps we failed to detect a significant correlation because wavelength is not the only key factor. The age of the subjects is also very important. In the next step, we attempted to assess the dependence of gamma stimulation effectiveness on age.

### 4.3. Dependence of the Effectiveness of γ-Stimulation on Age

We observed a weak trend towards decreased effectiveness of gamma stimulation with increasing age in mice (Figure 8). In humans, however, the effectiveness of gamma stimulation decreased only very slightly with age.

We next constructed 3D maps to determine the combined effects of age and wavelength on the efficacy of γ-light stimulation in humans (Figure 9a) and mice (Figure 9b).

The analyses performed showed significant differences in the distribution patterns of γ-stimulation efficiency, which depended on both age and wavelength. For mice, a fairly uniform region of high efficiency was found at frequencies of 500–600 nm and at an age of 4–5 months (see Figure 9). We believe that this high homogeneity is due to the consistency of the experimental studies and the methodological conditions. There are also additional regions of high efficiency. The first of these is found at an increased age (6 months) and at short wavelengths (below 500 nm). The second is at 2 months and in the infrared (≥1000 nm) range.

The following trends were detected in the case of humans (Figure 9). Firstly, there was a slight increase in the effective wavelength, from 500–600 nm to 600–700 nm. This is consistent with literature data on the increased effectiveness of gamma stimulation with increasing wavelength. Secondly, there is an increase in the effectiveness of gamma stimulation with decreasing age. This can be explained by increased brain plasticity in young people. Thirdly, there is a shift in effective wavelengths towards the short-wave region at the age of 40–50 years. This trend has not yet been clearly explained. One possible explanation is that it is a consequence of the complex nature of age-related changes in color perception [76]. In particular, sensitivity to green-blue, blue-green, and green colors decreases relatively quickly with age. However, the sensitivity of cones in the blue-violet range decreases more slowly with age [77]. It is possible that the shift in effective wavelengths towards the blue-violet region in middle age is due to the preservation of visual sensitivity in this range compared to others.

To assess the statistical significance of the observed dependencies of the effect on age and wavelength, we calculated z-scores for each effect value. According to our calculations, for mice, the log_10_ threshold for effects that differed significantly in z-score was 2.1. For humans, this value was slightly higher at 2.4. In both cases, the orange and red colors on the 3D maps correspond to areas with a z-score ≥ 2 (*p* < 0.05). Consequently, the main patterns are statistically significant.

Overall, the more complex relationship between frequency, age and efficiency in humans compared to mice may be due to humans’ longer lifespan.

### 4.4. Dependence of the Efficiency of γ-Stimulation on the Frequency of Stimulating Action

By definition, gamma oscillations have a frequency of 30–60 Hz. Therefore, the majority of authors have assumed that γ-rhythm stimulation should be performed at this frequency. However, resonance phenomena, harmonic generation and other effects are also possible, meaning that the enhancement of γ-rhythm generation by rhythmic stimuli with frequencies outside of the 30–60 Hz range cannot be ruled out. The vast majority of analyzed results were obtained using a periodic stimulus at 40 Hz: over 90% and 60% of studies for mice and humans, respectively (see Figure 10a,b). Frequencies below 40 Hz (usually 10–20 Hz) and, in isolated cases, above 40 Hz (60–80 Hz) were used much less frequently. Studies on humans revealed a wider range of effective frequencies, particularly in the low-frequency region. It is possible that the limited range of effective frequencies in mice is due to the methodological features of animal experiments.

However, although many studies in mice investigated frequencies of 10 and 20 Hz, statistically significant results were not obtained. Therefore, these results are likely due to differences in species between mice and humans. Our results for humans are consistent with previous studies. In several studies, gamma oscillations were induced by stimulation at lower frequencies. Stimulation at any of the frequencies 10, 20, 40 or 80 Hz activates EEG rhythms across the entire frequency range. When stimulation occurs at 40 Hz, a response is recorded for 10, 20 and 80 Hz simultaneously. Stimulation at 20 Hz produces a response for both 10 and 40 Hz. Furthermore, stimulation at any frequency leads to the same doubling effect in both directions. This process is very similar to generating the second and subsequent even harmonics, which often occurs in periodic processes. Harmonic oscillations can also be generated in the EEG, as described in detail in the literature [54,78]. Furthermore, global harmonic synchronization between rhythms and the influence of heart rate and other systemic processes in the human body have been described [78]. Firstly, the generation of harmonics is possible. This phenomenon was first described using human EEG rhythms a long time ago. Secondly, rhythms do not exist in isolation in the brain; changes in the frequency response of one rhythm can cause changes in the frequency response of another. For instance, an alteration in the amplitude of the alpha rhythm can result in a change in the amplitude of the γ-rhythm [79].

Due to the limited and highly specific frequency ranges used for gamma stimulation, we did not analyze the relationship between gamma stimulation effectiveness and frequency separately. However, considering that the gamma rhythm frequency response can change with age, we decided to analyze the ‘frequency–age–effectiveness’ relationship (see Figure 10f). For mice, a fairly uniform pattern of increase in the effective γ-stimulation frequency from 10–40 Hz to 40–60 Hz with increasing age was obtained (Figure 10). Three regions of interest have been ground for human:

The first is a wide frequency range of 40–60 Hz and an age range of 15–30 years. This can also include scattered areas with frequencies of 10–30 Hz and ages of 20–40 years. This pattern is consistent with the literature on age-related changes in the central frequency of γ-rhythms in humans [72]. The presence of subgroups responding to high- or low-frequency signals can be explained by the presence of two γ-rhythms, one with fast frequencies of 45–70 Hz and one with slow frequencies of 25–45 Hz [80]. It is likely that the expression of fast and slow gamma rhythms varies between individuals. Individual differences could be reflected in the distribution of young and middle-aged people according to their effective stimulation frequencies.

The second region is narrow, with a central frequency of ~10 Hz and an age of ~10 years. The presence of this region is consistent with data on age-related changes in the expression and responsiveness of the gamma rhythm to periodic stimulation. At school age (7–12 years), there is an increase in the effective central frequency [81]. It is likely that, at younger ages, sensitivity to frequencies below 40 Hz, including 10 Hz, predominates. In general, data on the effectiveness of gamma stimulation in children is relatively scarce, making it difficult to draw conclusions about this subgroup.

The third is a narrowly localized region with a central frequency of around 40 Hz and an age of around 60 years. It is noteworthy that a ‘dip’ appears in the frequency range of 45–80 Hz for people over 60 years of age. This is consistent with data on the age-related decrease in GABA concentration in visual, sensorimotor, frontal and prefrontal cortex areas [82,83,84,85,86,87]. A decrease in the central stimulating frequency to ≤36 Hz has also been widely described in the literature [80,88]. The position of the third region in the 30–40 Hz range is partially consistent with these data.

To assess the statistical significance of the observed effect dependences on age and wavelength, we calculated z-score values for each effect modules value (as described above). In this case, z-scores > 2 were achieved for log10 effect sizes ≥ 1.95 (red) and ≥2.95 (yellow, orange, and red) for mice and humans, respectively. Therefore, the main relationships and patterns, shown in Figure 9, are statistically significant.

### 4.5. Dependence of the Efficiency of γ-Stimulation on the Target

In addition to investigating the effectiveness of γ-stimulation in relation to stimulus characteristics (frequency, type and color), we also examined which responses γ-stimulation enhances more or less strongly. First, we chose to study the sensitivity of different levels of organization in living things. γ-stimulation: molecular, cellular and organismal levels. We also identified behavioral and cognitive responses, as these are of particular interest. Overall, that γ-stimulation was most effective in influencing organismal responses (see Figure 11). These responses included changes in cerebral blood flow velocity, frequency response and EEG rhythm coherence. However, γ-stimulation was less effective in influencing behavioral responses and cognitive and behavioral test performance. This highlights one of the main issues with this approach: the loose connection between electrical changes recorded in the brain and actual changes in subjects’ quality of life.

Our results suggest that relying solely on EEG frequency response and rhythm synchronization assessments, without conducting additional cognitive testing, could produce false positive results. To obtain the most reliable results and draw adequate conclusions, we recommend that future studies adhere to the following research design. First, use EEG, the most sensitive method, to test a wide range of gamma stimulation conditions (frequency, amplitude, signal shape, wavelength for light or frequency for sound, etc.) and select several optimal ones. Then, conduct cognitive or behavioral studies to validate the EEG results and select an effective stimulation protocol.

No differences were observed between changes at molecular and organismal levels. This may indicate a strong link between molecular signaling processes and manifestations at the organismal level, such as EEG and microcirculation. The effects of gamma stimulation are less pronounced at the cellular level than at the organismal level. This may be due to the greater plasticity of the physiological responses studied (e.g., EEG and vascular response) compared to cellular-level processes (which require time for signaling and functional proteins to be expressed) [89,90].

### 4.6. Dependence of the Effectiveness of γ-Stimulation on Pathology

Next, we evaluated whether the effectiveness of γ-stimulation differs between healthy subjects and individuals with pathological conditions. Based on the analyzed data, we selected the following groups: healthy subjects; patients with Alzheimer’s disease (AD); AD model animals; a stroke model; and patients with insomnia or other mental disorders. All analyzed literature data were divided by species: mice or humans. Overall, γ-stimulation was found to be equally effective in healthy humans and mice (see Figure 12). This suggests that animal models are suitable for fundamental research into the effects of gamma rhythm modulation. When considering each species separately, the effectiveness of γ-stimulation was found to be the same for healthy animals or subjects as for pathological cases. Animals with AD or a stroke, as well as intact animals, responded equally effectively to γ-stimulation. In patients with AD, insomnia and dementia, the effectiveness of gamma stimulation was comparable to that observed in healthy volunteers. This result further confirms the validity of the γ-stimulation approach in correcting pathological conditions.

However, there is a caveat. Significant differences in effectiveness are observed when evaluating the response to γ-stimulation in mice and humans with AD. Mice with AD showed a more pronounced response to γ-stimulation than patients with the disease did. Therefore, caution and accuracy are required when extrapolating data obtained from the AD mouse model to patients with the disease. Perhaps differences in sensitivity to γ-stimulation between species have slowed down research in this area. To obtain results that can be reliably transferred to humans, it is likely that additional pathology models must be created and/or identified. These new models should respond to γ-stimulation in a manner similar to that observed in real patients.

## 5. Mechanisms of γ-Stimulation Action

In order to understand the mechanisms of γ-stimulation, it is necessary to consider the external signal pathway, synchronization and subsequent effects in sequence. We will examine synchronization in detail, using visual stimulation as an example.

### 5.1. Visual Stimulation

The visual analyzer plays a direct role in the brain’s γ-activity. Electrophysiological data obtained from macaques showed that induced gamma synchronization between the primary visual cortex (V1) and the higher visual cortex (V4) facilitated the transmission of sensory signals for motor reactions, reducing reaction time [91]. It is well established that rhythms of electrical brain activity can differ in both frequency and localization [92]. The discovery of three types of narrow-band gamma rhythm in the primary visual cortex (V1) contributes to a deeper understanding of how information is processed by gamma oscillations. These three rhythms, known as low-frequency (25–40 Hz), mid-frequency (40–65 Hz) and high-frequency (65–85 Hz) γ-rhythms, process signals with different spatial frequencies and therefore convey distinct aspects of visual information in response to the original stimulus. However, they are actually generated by different neural networks [93]. In the field of multisensory cross-talk, it is widely believed that cross-modal matching of sensory signals depends on direct interactions between sensory cortical areas. Studies of human EEG with γ-stimulation (40 Hz) have confirmed that synchronized γ-oscillations in the cerebral cortex modulate multisensory communication during a visual–tactile matching task [94].

It has been demonstrated that brighter light results in greater rhythm synchronization [72,95,96]. Higher-amplitude light energy can cause more significant changes in the electrochemical properties of retinal photoreceptors and nerve conduction in the visual pathways [97].

Additionally, stimulation with longer waves elicited a stronger response than short-wave stimulation [98,99]. The cones in the retina are responsible for color vision, and in humans, cones sensitive to long wavelengths are denser than those sensitive to medium and short wavelengths [100,101]. Red light primarily excites medium- and long-wavelength cones (M- and L-cones), whereas white light also activates short-wavelength cones (S-cones). Studies have shown that stimulating L-cones strongly enhances the gamma rhythm in the visual cortex [102], whereas stimulating S-cones with shorter wavelengths fails to do so [103,104].

In practice, longer wavelengths are generally more effective at stimulating γ-rhythms [96]. However, the difference between white and red light flashes for young adults is insignificant. Moreover, a study involving older adults revealed no differences [72]. The lack of difference in sensitivity to white and red stimulation in older adults can be explained by two factors. Firstly, the spectral intensity of the white light used in this study consists mainly of longer wavelengths. Secondly, the age-related decrease in lens transparency reduces transmittance for shorter wavelengths [105,106]. These factors result in a decreased proportion of shorter wavelengths in white light. These factors may have influenced the results by reducing the difference in the effect of red and white light on γ-stimulation. Consequently, white light (peaking at 612 nm), which has a longer wavelength, may be as effective as red light (peaking at 614 nm) for γ-synchronization in older adults [15].

Visual γ-stimulation therefore primarily activates medium-wavelength (M) cones and long-wavelength (L) cones, meaning that longer-wavelength radiation has a greater effect on the γ-rhythm. This effect also depends directly on light intensity.

### 5.2. Modulation of GABAergic Interneuron Activity

The main type of neuron in the neocortex is the pyramidal neuron, accounting for 80–90% of all neurons. Those activated by gamma-aminobutyric acid (GABA) play a special role. GABAergic interneurons constitute 10–20% of the total number of neurons in the hippocampus and neocortex, and are characterized by high heterogeneity [15,67]. Extensive subclasses of GABAergic interneurons have been identified in both the hippocampus and the neocortex based on the expression of biomarkers such as parvalbumin (PV+), somatostatin (SST+) and vasoactive intestinal peptide (VIP+). The main source of excitation of inhibitory neurons in the cerebral cortex is the thalamus [107]. Typically, GABAergic neurons receive the greatest number of signals from thalamic areas that are most functionally connected to their own area of the cerebral cortex. For instance, in the primary auditory and visual cortices, inputs from the medial and lateral geniculate bodies of the thalamus, respectively, elicit excitatory postsynaptic currents in PV+, SST+ and VIP+ interneurons [107,108]. These areas are involved in visual and auditory γ-stimulation.

In this way, the entire synchronization pathway of gamma oscillations can be traced using external stimuli. Sensory stimuli and behavioral responses are reflected in the thalamus, which subsequently modulates the activity of interneurons. This establishes a dynamic functional connection between inhibitory interneurons and pyramidal cells, enabling the latter’s activity to be controlled flexibly in real time. This connection ensures the short-term plasticity of synaptic inhibition and consequently endogenous γ-oscillations in response to external sensory input (see Figure 13).

Furthermore, PV+ interneurons are responsible for generating γ-oscillations, whereas SST+ interneurons generate θ-oscillations. These two types of oscillation typically exist together as θ-nested gamma oscillations, where θ-rhythms are responsible for the temporal characteristics of the transmission of information in packets, and nested γ-rhythms are responsible for the transmission of information itself [109]. This ensures synchrony between brain regions, which is important for working memory, information retrieval, cognitive integration and processing information. Therefore, the correct activity of inhibitory interneurons is essential for forming certain brain states and ensuring healthy cognitive functioning, and this activity can be influenced by external stimuli.

At a higher organizational level, interneurons combine to form specific networks. Gamma rhythms, which are generated by interconnected GABAergic inhibitory interneurons, regulate the global balance of excitation and inhibition in the visual cortex [110,111,112]. Synchronous oscillations in the gamma range are generated by two types of interneuron network: PING (excitatory–inhibitory networks of neurons) and ING (purely inhibitory populations of neurons) [113,114]. The power and frequency of γ-oscillations are modulated by stimulus characteristics (e.g., direction, speed, contrast) [16,115].

The frequency of γ-oscillations is determined primarily by inhibitory postsynaptic currents induced by GABA receptors. In the ING network, interneurons mutually inhibit each other via GABA receptors (Figure 2) to quickly achieve synchrony in phase 0. During the PING mechanism, pyramidal cells induce rapid excitation of interneurons via AMPA receptors, which in turn ensures inhibition via GABA receptors and triggers γ-oscillations.

Meanwhile, brain properties such as resting GABA levels and the size of the affected cortical area also play a role. The relationship between the frequency used for visual synchronization and the excitability of GABAergic neurons is of interest, as this depends on the level of GABA production in the human body. It has been noted that, for the most effective synchronization with γ-rhythms in mature individuals, a frequency of about 32–34 Hz is necessary, which is less than 40 Hz [72]. This age-related decrease in the optimal flicker frequency may be due to a corresponding decrease in the central frequency of γ-rhythms in older individuals. The central frequency is the frequency at which power changes most strongly in response to external visual stimulation. Notably, even within a single area (the hippocampus), the presence of two γ-rhythms with distinct frequencies is possible: one fast (45–70 Hz) and one slow (25–45 Hz) [80]. These rhythms can be stimulated differently by the same stimulus, and their response depends on stimulus orientation, contrast, drift velocity and amplitude [80]. With increasing age, the amplitude of fast gamma oscillations decreases more significantly than that of slow ones [88]. According to some studies, the central frequency of the stimulus increases monotonically with the intensity of the visual stimulus: visual contrast [116,117,118], movement speed [16,115] and positively correlates with the GABA level in the visual cortex [116,118]. Moreover, the central frequency of the stimulus increases as the tonic excitability of GABAergic inhibitory interneurons increases [119,120]. The central frequency gradually decreases with age (by 0.16 Hz per year in the high γ-activity range of 36 Hz and above and by 0.08 Hz per year in the low γ-activity range below 36 Hz) [80,88]. It has been shown that GABA levels decrease in the visual, sensorimotor, frontal and prefrontal areas of the cortex with age in humans [82,83], which may impair inhibitory intracortical circuits [84,85,86,87].

Studies [120,121] have documented age-related differences in central frequency during visual γ-stimulation in humans. This age-related decrease in central frequency may be associated with a reduction in the excitability of GABAergic inhibitory interneurons, which is caused by a decrease in GABA production [83,122]. Thus, the effect of γ-stimulation is realized through GABAergic interneurons and depends on the level of GABA production, which determines the excitability of these neurons and the central frequency of the response to external stimuli. Moreover, as people age, the central frequency may decrease, meaning that the flickering rhythm for effective γ-stimulation must be selected on an individual basis. This rhythm may ultimately differ from 40 Hz.

### 5.3. Acetylcholine Mechanism of γ-Oscillation Activation

It has been established that including cognitive tasks such as attention, memorisation, counting and searching for inconsistencies in gamma stimulation sessions promotes the spread of the gamma rhythm to additional areas of the brain, including the frontal and deep lobes such as the hippocampus [123,124]. Additionally, coherence and cortico-cortical interactions between different brain regions are enhanced [125,126,127]. These effects can be explained by the fact that, in addition to GABAergic transmission (used in both ING and PING, see above), the PING network with asynchronous excitation can generate γ-oscillations in response to impulses of acetylcholine (ACh) through muscarinic ACh receptors (mAChRs). This ensures the network can adapt dynamically during tasks that require high levels of concentration [128]. Signal detection in behavioral attention tasks depends on cholinergic-driven γ-oscillations in the frontal cortex, to which both mAChRs and nAChRs make a significant contribution [129]. Therefore, cognitive attention tasks enable the involvement of additional neural populations (PING) and the spread of γ-oscillations induced by external stimuli to be much broader and deeper.

In addition, N-Methyl-D-aspartic acid (NMDA) receptors are involved in the coherence and amplification effects of the synchronized γ-signal. When external stimuli induce γ-synchronization in the network, functional NMDA receptors are activated in the excitatory feedback synapses between CA1 pyramidal cells and parvalbumin-positive (PV+) interneurons. This improves and stabilizes neuronal assemblies [130,131]. NMDA receptors in PV+ cells generate a relatively slow postsynaptic current and are involved in controlling spontaneous and induced γ-oscillations. This explains why NMDA receptor blockers (ketamine, MK-801, phencyclidine, etc.) affect γ-oscillations [130,132]. In addition, the involvement of sodium channels Nav1.1 (Figure 13), a subunit of voltage-gated sodium channels, is important. t is expressed in interneurons following neddylation to maintain the excitability and the excitatory/inhibitory balance of GABAergic interneurons [131,133,134]. Nav1.1 has been observed to promote behavior-dependent γ-oscillations [134].

Thus, to expand γ-synchronization areas, it is important to use a cognitive task (see below) alongside visual or auditory stimulation. This activates a parallel mechanism associated with γ-oscillation activation and acetylcholine. Voltage-gated sodium channels are also involved in this process. NMDA receptors stabilize and enhance the resulting neural coalitions (assemblies), ultimately achieving interareal coherence.

### 5.4. Modulation of Neural Network Activity

The current neuroscientific paradigm is that cognitive tasks are not performed by discrete brain regions operating in isolation, but by networks consisting of several such regions that are said to be ‘functionally connected’ [135,136,137].

As mentioned above, gamma oscillations coordinate communication between different parts of the cortex, ensuring the brain’s temporal and spatial connectivity [11,138]. The set of interconnected brain regions forming large-scale networks varies depending on the cognitive function [139,140]. Large-scale brain networks are identified by their functions and grouped into self-organizing coalitions. The number and composition of these coalitions vary depending on the algorithm and parameters of the problem being solved [135,141,142,143].

Surprisingly, despite the significant importance of understanding that the brain functions as a network and that gamma oscillations mediate communication between different brain regions, we found no studies in the literature on the effects of gamma stimulation on the activity of known neural networks. The few literature sources we were able to find demonstrating the possibility of such an effect do not yet provide a comprehensive picture and are quite sparse. We would like to draw the attention of researchers to this extremely interesting area of research. Here we present information on three of the most studied neural networks and our findings on their relationship with γ-stimulation.

#### 5.4.1. Basic Modes of Brain Function and γ-Stimulation

Abnormalities in activity in various networks are associated with neuropsychiatric disorders such as depression, Alzheimer’s disease, autism spectrum disorders, schizophrenia, ADHD, and bipolar disorder [135,136,144,145,146]. Numerous functional magnetic resonance imaging (fMRI) studies have shown that Alzheimer’s disease is associated with atypical patterns of functional connectivity in large-scale brain networks [147,148,149,150,151,152,153,154,155]. During the default mode, various brain centers remain functionally connected, and the characteristics of network connectivity in this state can depend on factors such as gender, age, motor asymmetry profile, overall EEG pattern, and the presence of neuronal pathologies [156]. In particular, the functional connectivity of the brain’s default mode network is altered in Alzheimer’s disease. It is weakened in the hippocampus and medial lumbo-parieto-occipital regions [157,158]. Disruption of the default mode network is associated with the deterioration of cognitive functions in elderly individuals [159]. Recently, researchers have paid more attention to inter-network connectivity patterns in the insula to better understand the pathological changes in the organization of brain function in Alzheimer’s disease [160,161,162].

The three largest-scale networks that are responsible for cognitive functions are the default mode network, the executive network and the salience network. Together, these form the so-called ‘triple network model’ of Menon [11,135,147,163]. General neuronal network and γ-stimulation effects are shown at Figure 14.

#### 5.4.2. The Default Mode Network

Default mode network (DMN), also known as the neural network of operational rest, passive mode of brain function network, is a network of interacting brain areas that becomes active when a person is not performing any task related to the outside world, but rather is inactive, resting or ‘immersed in themselves’ [164].

The DMN includes several anatomically separate but functionally interconnected areas of the brain: the ventromedial prefrontal cortex; the dorsomedial prefrontal cortex; the lateral parietal cortex; and the posterior cingulate cortex with adjacent parts of the precuneus [165,166,167]. The DMN often includes the entorhinal cortex, which is associated with the hippocampus [168]. Significant connections have been shown between the DMN and the hippocampus and the thalamus, but not with the basal nuclei [169].

##### Functions of the DMN

Classically, the functions of the DMN are the processing of information about the internal world, including thoughts, memories, and emotions [170]. It also integrates past, present, and future [171,172]. It integrates information about the past retrieved from memory, signals perceived by the senses, and plans and images of the future. The DMN brings them together and facilitates understanding of what is happening in the current moment. It connects the points of the human life timeline and performs the following functions:Maintaining flexibility in thinking is the ability to switch from one task to another [173,174].Supporting autobiographical memory and integrating memories. It establishes deep connections with both the inner self and the surrounding world [175,176].Social thinking: thinking about the intentions and feelings of others [177,178].Supporting creative self-expression. The DMN has the ability to establish connections between individual areas of the brain, thus enabling unique associations and the development of a person’s individuality. This enables spontaneous and spontaneous action [179,180,181].Promotes clearer manifestation of vague memories. The DMN engages memory beyond the boundaries of attention, extracting information that cannot be retrieved by other means. [182,183].

Dysfunction of DMN or abnormal changes in the functional connections between brain regions have been found in people with depression, bipolar disorder, schizophrenia and other mental health conditions [184]. For example, altered activity and disrupted communication in the frontal cortex, hippocampus, and dorsomedial prefrontal cortex have been observed in depression [185,186,187,188]. Changes to these structures have been associated with the onset or intensification of rumination, the deterioration of cognitive function and the impaired processing of emotional stimuli. In patients with schizophrenia, hyperactivation of the DMN has been observed, which interferes with the processing of information from external stimuli [189,190].

##### Effects of γ-Stimulation on the Functioning of the DMN

Chronic (8 weeks daily) audiovisual γ-stimulation increased functional connectivity between the posterior cingulate cortex and precuneus (leading regions of the DMN) in patients with AD [191]. A decrease in cortical atrophy and an improvement in DMN function were also observed in patients with mild AD with long-term (3 months) audiovisual stimulation [192]. Results obtained using auditory stimulation at a frequency of 40 Hz showed an increase in neuronal activity in the parietal and prefrontal regions where the DMN is located [193,194]. In another study, audiovisual stimulation (3 months of daily stimulation) did not lead to changes in connectivity in the DMN, but significantly increased the average functional connectivity in the motor cortex of patients with mild AD [62].

#### 5.4.3. Central Executive Network

##### Central Executive Network Functions

This neural network provides a solution of task that requires concentration, planning or problem-solving needs to be completed [195,196]. It is in a reciprocal relationship with the DMN. It consists of the dorsolateral prefrontal cortex and the anterior cingulate cortex.

Some functions of the executive network include:Concentration and attention: maintaining focus on the task.Working memory involves retaining and manipulating information in the mind.Planning and control involves developing a strategy of actions and controlling their implementation.Inhibiting impulses and preventing rash actions.

##### Effects of γ-Stimulation on Central Executive Network

It has been demonstrated that the activation of working memory, one of the primary functions of the executive network, enhances the amplitude of gamma-band oscillations and strengthens their coherence across various regions of the cerebral cortex [197,198]. Furthermore, transcranial electrical stimulation of two distant areas within the gamma band has been shown to enhance working memory task performance [199]. These results suggest that γ-band coherence probably plays a significant role in maintaining working memory.

#### 5.4.4. Salience Network Executive Network

##### Salience Network Functions

This network comprises several structures, including the bilateral anterior insula, the dorsal anterior cingulate cortex, and three subcortical structures: the ventral striatum, the substantia nigra, and the ventral tegmental area [200,201]. This network plays a pivotal role in monitoring the significance of both external stimuli and internal brain activity [202,203,204]. In particular, salience network helps direct attention by identifying important biological and cognitive events [205]. Salience network regulates the activity of the aforementioned two networks, activating them depending on the environment.

##### Effects of γ-Stimulation on the Salience Network

Studies have shown that visual stimulation at 40 Hz activates the cortex, the hippocampus and the insular cortex, which is part of the salience network [123]. Thus, the enhancement of cognitive abilities during gamma synchronization is realized through modulation of the activity of various brain networks responsible for different cognitive functions, such as DMN, the central executive network, etc.

### 5.5. Activation of Neurogenesis

Long-term exposure to 40 Hz flickering light (one hour per day for 30 days) has been shown to significantly improve spatial learning and neurogenesis in the dentate gyrus of the mouse hippocampus, with no detrimental behavioral side effects observed [206]. As discussed above, this effect is based on the activation of PV+ interneurons and the GABAergic support of immature neurons. This demonstrates the long-term benefits of this treatment for neurological diseases. Mechanistically, the stimulation did not alter regional microvascular blood flow, but it did significantly increase PV+ interneuron excitability, GABA levels and inhibitory transmission in the dentate gyrus of the hippocampus. Blocking GABA receptors abolished the improvements in spatial learning and neurogenesis. These data demonstrate that long-term exposure to 40 Hz flickering light improves spatial learning through PV-dependent adult neurogenesis, a process that requires elevated GABA levels to support neurogenesis in adult mice.

In adult mice, neurogenesis predominantly occurs in the granule cell region of the hippocampus and the subventricular zone of the lateral ventricles. The formation of new neurons in the granule cell region during adulthood is critical for learning and memory formation. Neurogenesis occurs predominantly in the granule cell region of the hippocampus and the subventricular zone of the lateral ventricles in the adult mouse brain. The formation of new neurons in the granule cell region during adulthood is critical for learning and memory formation [207,208,209,210].

The precise regulation of neurogenesis in the hippocampus largely depends on localized neural circuits that integrate individual experience, neuronal activity, and the regulatory mechanisms of neurogenesis [211]. Within these circuits, two main types of GABAergic interneuron play a vital role in supporting the development of new neurons. The first are rapidly excitable PV+ interneurons, and the second are SST+ interneurons. Both types of interneuron inhibit granule cell dendrites. PV interneurons are typically located closer to the granule cell layer, providing lateral and reciprocal inhibition and promoting continuous GABA release [212,213]. Synaptic connections between PV interneurons and neonatal neurons can be observed as early as seven days after birth, i.e., earlier than connections established by SST+ interneurons [213]. Previous studies have shown that optogenetic inhibition of PV+ interneurons, but not SST+ interneurons, effectively suppresses neurogenesis in the granule cell region [214].

The activation of PV+ interneurons can serve as a sensor of circuit activity, providing local signals that activate various components in the hippocampal neurogenic niche. Notably, GABA influences immature neurons by increasing their resting potential through its side effects and subsequent synaptic release, making them more receptive to action potentials [215,216,217,218]. This excitatory effect of the inhibitory neurotransmitter GABA on neonatal neurons is explained by the high expression of NKCC1 chloride channels, which depolarize the chloride ion equilibrium potential in neonatal neurons, particularly during the first three weeks after birth [213,219,220].

In order for inhibitory transmission to support neuronal maturation in the dentate gyrus of adult mice, GABA release must be enhanced by the activation of interneurons. This can be achieved through prolonged exposure to 40 Hz flickering light [206]. Thus, one of the key effects of PV+ GABAergic neuron activation is the stimulation of neurogenesis in the hippocampus, which is certainly beneficial for the long-term positive effects of γ-synchroenzyme.

### 5.6. Adenosine Hypothesis

An alternative mechanism by which γ-stimulation exerts its effects is the enhancement of adenosine production [221]. The main sources of this production are glutamatergic and GABAergic neurons. In this case, however, astrocyte activity does not play a decisive role in the production of extracellular adenosine. Additionally, an increase in the calcium signal was observed in the V1 area of mice. Furthermore, the calcium channel blocker feldopidine (10 mg/kg, intraperitoneally) almost completely eliminated the increase in extracellular adenosine levels induced by 40 Hz flicker, as compared with the placebo group. This is an important consideration when using γ-stimulation in humans, since many drugs affect calcium channels and can therefore reduce the effectiveness of the stimulation.

Light-induced neural activity (e.g., spiking activity or γ-oscillations) is particularly energy-consuming and requires coordinated mitochondria. It also results in intense oxygen consumption and hemodynamic changes. This is evidenced by the dependence of signal frequency responses in the cerebral cortex on blood oxygen concentration [222,223]. As adenosine is the main signaling molecule resulting from increased energy expenditure [224,225,226], it is highly likely that light-induced neural activity represents a form of energy-consuming neural stimulation resulting in rapid adenosine production. Extracellular adenosine, which is closely associated with energy metabolism, can act as a modulator of neurotransmission and synaptic plasticity, as well as regulating metabolic homeostasis, motility, proliferation and vasodilation in the brain [227,228,229].

Thus, prolonged exposure to a single frequency (40 Hz) is extremely energy-consuming and activates energy metabolism with the production of significant amounts of adenosine, which has significant biological effects (hypnotic effect, increased neuroplasticity, modulation of neurotransmission, vasodilation, etc.) [230].

### 5.7. Effect of γ-Stimulation on Microglia and Inflammation

The occurrence of abnormal γ-oscillations alongside neuroinflammation indicates disruption to the microenvironment of the CNS neural network. In the CA3 region of the hippocampus, aged *nfkb−/−* mice, which exhibit low-grade sterile inflammation with aging, demonstrate reduced γ-frequency oscillation power [231]. Various levels of neural network dysfunction are caused by microglial activation, primarily due to the excessive release of reactive oxygen species and nitric oxide via Toll-like receptors (TLRs), rather than proinflammatory cytokines [232]. IFN-γ-activated microglia enhance the expression of inducible nitric oxide synthase (iNOS) and subsequently produce large amounts of nitric oxide, leading to a decrease in γ-frequency oscillations and an increase in neuronal death in hippocampal slices in situ [233]. Combined with lipopolysaccharide and interferon-γ stimulation, hippocampal cells exhibit a significant loss of PV+ neurons and reduced or absent γ oscillations [233]. Rodent studies and clinical trials have demonstrated that anti-inflammatory therapy can restore gamma rhythms and enhance cognitive function [234,235]. These results highlight the sensitivity of γ oscillations to disruption of CNS homeostasis and the role of microglia, the brain’s resident immune cells that provide dynamic surveillance, in maintaining γ oscillations in disorders associated with various microglial functions, such as the immune response, reactive oxygen species production and synaptic remodeling.

Visual stimulation at 40 Hz has been shown to significantly reduce neuronal loss in the visual cortex (V1), the CA1 region of the hippocampus, the somatosensory cortex and the cingulate cortex in mice with neurodegeneration. It has also been shown to reduce the loss of CA1 excitatory neurons in mice with ischemic stroke. However, the mechanism of this effect is not fully understood. Studies have shown that gamma stimulation suppresses the expression of inflammatory genes and reduces DNA damage. A γ-stimulation has also been shown to protect mice with P301S and CK-p25 mutations from severe neuronal loss and brain atrophy [62]. Interestingly, restoration of neuronal survival following ischemia does not appear to depend on changes in cerebral blood flow or the response of microglia, suggesting that visual stimulation may directly affect neurons.

The immune function of the brain is generally attributed to microglia, the brain’s primary phagocytes. This is because they engulf pathogens and release cytokines and other extracellular signaling molecules that maintain neuronal health [236,237,238]. In a healthy adult brain, the processes of microglia move dynamically to explore the surrounding synapses. They acquire an elongated shape in order to scan the environment and monitor neuronal activity (hyperbranching). Upon detecting pathogens, the microglia rapidly changes their phenotype. They pass through intermediate stages where their processes shorten and their bodies enlarge (hypobranching), transforming into an amoeboid form that may lack processes entirely [237].

Depending on the nature of the pathological condition and the resulting changes to the microenvironment, different types of microglia can be activated [239]. The two extreme types are polarized M1 and M2 cells, as described for peripheral macrophages [240]. M1 microglia play a vital role in combating infection and injury, the majority of factors they secrete are toxic to neuronal cell cultures. Conversely, in the presence of IL-4, IL-13 or IL-10, microglia differentiate into M2 or alternative microglia. These are characterized by the expression of IL-10, heparin-binding lectin (Ym1), the cysteine-rich protein FIZZ1 and arginase 1 [241]. Although M1 microglia plays an important role in combating infection and injury, most factors secreted by M1 microglia are toxic to neuronal cell cultures. On the other hand, in the presence of IL-4, IL-13, or IL-10, microglia differentiate into M2 microglia or alternative microglia, which are characterized by the expression of IL-10, heparin-binding lectin (Ym1), the cysteine-rich protein FIZZ-1, and arginase 1 [241]. The anti-inflammatory M2 phenotype exhibits neuroprotective properties and is involved in resolving inflammation, phagocytosis, and tissue repair.

The M1/M2 microglial activation paradigm is being studied more and more in the context of neurodegenerative and neurological diseases, where an imbalance towards the M1 form is observed [242,243,244]. Thus, normalizing the imbalance between M1 and M2 differentiation has been proposed as a therapeutic target for treating CNS diseases associated with neuroinflammation. In this context, IL-10 combined with IL-13 has been shown to enhance the secretion of activin-A by microglia. Activin-A is a neuroprotective member of the TGF-β superfamily that promotes oligodendrocyte differentiation [245].

It was found that flickering at 20 and 40 Hz leads to different patterns of cytokine expression in healthy mice [67]. It was previously established that microglia change their morphology within a few minutes [246]. One hour of flickering at 40 or 20 Hz was found to alter microglial morphology consistent with their different functional roles, and also alter the transcription of genes controlling cytokine expression [67,236]. Flickering at 40 Hz caused a significant increase in somatic area (hypobranching) compared to 20 Hz, while 20 Hz caused a significant increase in process length and branching compared to 40 Hz. Thus, 40 Hz stimulation results in a more amoeboid morphology with fewer branches, consistent with ‘engulfing’ microglia phenotypes, while 20 Hz stimulation results in a more branched morphology with longer processes, consistent with ‘surveilling’ microglia phenotypes [37,236,247].

Flickering at 40 Hz increased the production of a wide range of cytokines. Some cytokines, including the anti-inflammatory cytokine IL-13 and the chemokine macrophage inflammatory protein-1β (MIP-1β) are secreted by microglia [248]. The γ-stimulation in microglia-deficient mice was found to increase tissue levels of cytokines such as macrophage colony-stimulating factor (M-CSF), which attracts microglia, and IL-10 with anti-inflammatory effects [67]. The authors conclude that these cytokines are produced by cells other than microglia. Microglia deficiency was induced using PLX3397 (an inhibitor of the colony-stimulating factor 1 (CSF1) receptor), which can kill macrophages [249]. The anti-inflammatory agent IL-10 on the one hand suppresses the production of TNF-α, but at the same time paradoxically slows the transformation of microglia from an activated phagocytic state to an intact one [250]. It can be assumed that this is the basis of the effect of γ-stimulation, which, on the one hand, reduces the severity of toxic inflammation, and on the other, activates the phagocytic activity of microglia (transferring it to the M2 state), facilitating the removal of amyloid plaques and tau protein.

IL-10 is a key cytokine that counteracts neuroinflammation. It suppresses the production of proinflammatory cytokines by microglia, thereby protecting astrocytes from excessive inflammation [251,252]. In neurons, signaling through the IL-10 receptor has been linked to increased cell survival and the regulation of neurogenesis in adults [253,254,255,256,257]. Therefore, IL-10 is a crucial mediator of intercellular interactions between microglia, astrocytes and neurons.

Notably, alongside the existing data suggesting that IL-10 regulates intercellular interactions within the CNS and immune system, there is a growing body of research indicating impaired IL-10 production or signaling in patients and animals with various neurological diseases, including neuropathic pain, multiple sclerosis, Alzheimer’s disease and Parkinson’s disease [255,258,259,260,261,262].

Furthermore, it was found that inhibiting nuclear factor kappa-light-chain-enhancer of activated B cells (NF-κB) or mitogen-activated protein kinases (MAPK) signaling pathways (which are responsible for synaptic and immune function) prior to exposure to flicker reduced cytokine expression; meanwhile, inhibiting NF-κB prevented the microglial response to 40 Hz flicker [67]. Taken together, these results suggest that flicker at different frequencies has different effects on microglia and cytokine transcription.

Another study found that exposing wild-type mice to γ-flickering light increased levels of cytokines such as IL-6 and IL-4, which enhanced microglial phagocytosis and increased the expression of chemokines, including macrophage colony-stimulating factor and interferon-gamma-induced monokines. This neuroimmune activation was mediated by γ-induced phosphorylation of proteins NF-κB and MAPK-dependent pathways in astrocytes [263,264,265].

The signaling molecules protein kinase R (PKR), c-Jun N-terminal kinase (JNK), and NF-κβ have been identified as likely participants in the regulation of TLR3-dependent IL-10 expression in astrocytes [266]. Therefore, it may be that astrocytes are able to realize the effects of γ-stimulation via the NF-κβ pathway mentioned earlier.

However, there is also an opposing point of view. In the context of Alzheimer’s disease, introducing IL-10 into the brains of transgenic mice with the amyloid precursor protein resulted in the accumulation of Aβ and a decrease in its phagocytosis by microglia [267]. The negative effect of IL-10 on Alzheimer’s disease was confirmed by the observation that the absence of IL-10 in transgenic mice with cerebral amyloidosis resulted in Aβ being phagocytized by activated microglia, leading to a reduction in Aβ levels in the mice’s brains and ultimately alleviating the disease’s symptoms to some extent [268]. Furthermore, the authors propose that a potential therapeutic strategy could involve restoring the balance of innate immunity in the brain by suppressing the action of key anti-inflammatory cytokines, such as IL-10, enabling the brain to return to a physiological state [268].

Phagocytic activity is enhanced. A reduction in the number of Aβ plaques in the visual cortex of 5XFAD mice exposed to γ-stimulation correlates with the morphological transformation of microglia [58]. Another study also found a reduction in the level of phosphorylated tau protein in mice with tauopathy with a similar transformation of microglia [62]. However, γ-oscillations induced by visual stimulation did not significantly affect the transition of microglia to a phagocytic state, the number of microglia, or markers of neuroinflammation in aged C57BL/6J mice [62]. Similarly, the response of microglia to gamma stimulation was limited in an animal model of ischemic stroke. This suggests that the effects of γ-stimulation on microglia may depend on health status or genetic predisposition [269]. This may be due to an age-related decrease in the excitability of GABAergic neurons and the level of GABA in the cortex [82,83,122].

Thus, γ-stimulation also has immunological effects. On the one hand, it reduces toxic inflammation, thereby preventing DNA damage and neuronal death. On the other hand, it activates the phagocytic activity of microglia, preventing them from returning to their original state for a prolonged period and helping to remove Aβ and tau protein. However, the extent to which these effects are observed may depend on the individual’s age and general health. For convenience, a summary table of all the data discussed in the article is provided below (Appendix A, Table A1).

## 6. Optimization of γ-Stimulation

Selection of individual parameters associated with biological age. The effectiveness of visual γ-stimulation is affected by: brightness intensity, color (wavelength), flicker frequency, time and duration of exposure [95,99,270,271,272,273]. Also, the optimal parameters of SMS for γ-synchronization in elderly people may differ from those in young people [88,274]. Age-related differences in γ-synchronization using light stimuli may be associated with age-related changes in the eyes and brain. With age, retinal illuminance becomes less effective due to a decrease in pupillary miosis and color discrimination. The decrease in illuminance associated with lens yellowing is not the same for all wavelengths, and most color defects in the elderly are of the blue-yellow type [275,276]. The central frequency of γ-rhythms gradually decreases with age [88].

Let us consider these parameters in turn. Optimal parameters of GENUS for γ-wave synchronization in humans may differ from those in mice. In humans, γ-waves synchronized by light flickering with a frequency ≤ 54 Hz propagate to the frontocentral region, while those synchronized by light flickering with a frequency > 54 Hz do not [277]. The spectral power of the involved γ-waves increased with increasing light intensity. The color (wavelength) of light can also influence γ-wave stimulation [95,278].

### 6.1. Visual γ-Stimulation in Young Adults

The optimal color, intensity and frequency of a flickering light stimulus for activating gamma waves differed between young adults (mean age 24 years) and more elderly persons [96]. When stimulating the brain with different colors (white, red, green and blue), red light was found to be the most effective in eliciting γ-waves, followed by white light. Light of higher intensity (400–700 cd/m^2^) elicited stronger γ-waves than light of lower intensity (10–100 cd/m^2^). The γ-waves were stronger and extended beyond the visual cortex when light flickered at a frequency of 34–38 Hz than when it flickered at a frequency of 40–50 Hz. Light with an intensity of 700 cd/m^2^ caused more severe side effects (eye fatigue, eye pain and glare) than light of other intensities. White light with a luminance of 400 cd/m^2^, flickering at a frequency of 34–38 Hz proved optimal for activating intrinsic γ-rhythms in young people.

The authors suggest that, although 40 Hz stimulation has been used in studies on AD mice [21,42,43], future human clinical studies should consider using 34–38 Hz stimulation instead, since 30 Hz light flickering has also been shown to prevent neurodegeneration of pyramidal neurons in the hippocampus of ischemic mouse models [47].

### 6.2. Visual γ-Stimulation in Older Adults

The optimal parameters of visual stimulation for synchronization with gamma rhythms in elderly people with normal cognitive function were studied at different brightness levels, colors and flicker frequencies [60].

The 700 cd/m^2^ flicker induced a stronger and more widespread γ-rhythm than the 400 cd/m^2^ flicker, and had no significant negative effect [60]. Furthermore, flicker at 32 Hz or 34 Hz induced a stronger and more widespread γ-rhythm than flicker at other higher frequencies. However, white and red flicker induced γ-rhythm equally in the brains of older adults.

Thus, older adults may tolerate brighter light better than younger adults, possibly due to age-related increases in miosis and lenticular cell senescence. For this reason, older adults may require brighter light stimulation to synchronize with the gamma rhythm, similar to younger adults. In patients with Alzheimer’s disease (AD), adverse effects from exposure to very low levels of light (VLS) of approximately 700 candela per square meter (cd/m^2^) were also shown to be minor [191].

The flicker frequency at which the strongest and most widespread γ-rhythm synchronized was 32 or 34 Hz in older adults and 36 or 38 Hz in young adults in a previous study [96]. The γ-rhythms synchronized to 32 or 34 Hz were approximately 1.2 times stronger at the Pz (center of head) site and 1.4 times stronger at the Fz (frontal) site than those synchronized to 40 Hz in older adults. Furthermore, γ-rhythms synchronized to 32 and 34 Hz were approximately 1.3 times stronger, covering approximately twice as many nodes as those synchronized to 40 Hz. In young adults, stimulation at 38 Hz synchronized γ-rhythms approximately 1.2 times more strongly in both Pz and Fz and covered approximately twice as many nodes, with a strength 2.8 times higher than at 40 Hz. These results indicate that the optimal flicker frequency for γ-rhythm synchronization in humans may differ significantly from that in mice and may also decrease with age.

### 6.3. The Problem of Stroboscopic Flicker and Its Solution

Using stroboscopic flicker, also known as temporally modulated light, for medical purposes is associated with certain difficulties due to its flickering nature. In human safety and feasibility trials, 28% of participants discontinued the therapy, which may indicate an issue with stroboscopic stimulation [279]. However, one study shows the safety of audiovisual gamma stimulation [191]. The side effects of stroboscopic light decrease as the temporal modulation frequency increases, until flickering is no longer perceived. The critical flicker fusion frequency is the frequency at which temporally modulated light ceases to be perceived as flickering [280]. It depends on the color and depth of the flicker modulation, as well as on the characteristics of the observer. For strobe lights with luminance of 100–500 lux, which were used in studies [24,30,37,58,59,62], the flicker fusion frequency is in the range of 55–70 Hz [280], that is, at least 15 Hz higher than the stimulation frequency of 40 Hz, making the flicker very noticeable to observers.

Heterochromatic (or simply chromatic) flicker replaces the “off” phase of stroboscopic flicker with a phase consisting of light of a different color, so that the light alternates between two different colors. Replacing stroboscopic flicker with chromatic flicker can reduce the critical flicker fusion frequency to 15–30 Hz (depending on luminance), improving neural oscillatory response and reducing potential side effects. This makes chromatic flicker a possible candidate for a 40 Hz stimulation paradigm in humans.

Spectral adjacent color pairs appear less flickering to the perceiver, so these results encourage further exploration of the limits of phase similarity that still elicit a 40 Hz response. Specifically, pairing a color at one extreme of the visible spectrum with another similar color may provide the best compromise between the perception of flicker and the magnitude of the steady-state visual evoked potentials (SVEP).

The study presented a new type of heterochromatic flicker based on spectral combinations of blue, cyan, green, light green, amber, and red colors (BCGLAR color space) [281]. The combination of amber and red flicker induced the highest SVEP, and combinations including blue and/or red colors consistently induced higher SVEP than combinations with only mid-spectrum colors.

The creation of an invisible spectral flicker (ISF) that induces neural synchronization at a frequency of 40 Hz and a spatial distribution similar to a 40 Hz stroboscopic light [282]. The type of ISF used in this study involves alternating two spectrally distinct light compositions, making the flickering imperceptible to the observer but creating the sensation of white light, similar to continuous light. The subtle flickering characteristic of ISF makes it similar to light therapy. It has already been recognized as an effective treatment for various mental disorders and dementia [283,284].

It must be acknowledged that the stroboscopic signal induced a significantly greater SVEP than ISF (approximately 4 times [282]); nevertheless, the resulting method allows for more comfortable stimulation and opens up the possibility of conducting future randomized, placebo-controlled clinical trials with an acceptable double-blind method due to the virtually imperceptible flicker, which is expected to significantly reduce discomfort compared to interventions based on stroboscopic flicker.

### 6.4. Increasing the Exposure Time by Stimulation During Sleep

As shown above, the results of γ-stimulation and the removal of neurotoxic molecules are inconclusive. The authors suggested that the limited duration of stimulation was a limiting factor [285]. They tested the feasibility of applying visual stimulation at a frequency of 40 Hz during sleep. This stimulation effectively induced γ-activity in neurons at the stimulation frequency without deteriorating sleep quality, thus confirming the feasibility of this approach. Furthermore, this approach could utilize glymphatic system activation to remove neurotoxic Aβ molecules from the brain. This neural mechanism can be exploited through stimulation at a frequency of 40 Hz during sleep [286,287]. Therefore, in addition to improving ease of use, stimulation at 40 Hz during sleep is likely to be more effective.

The well-established connection between the glymphatic system, dementia and sleep has already resulted in some fascinating developments in photobiomodulation therapy [288,289,290,291]. Photobiomodulation involves applying near-infrared light to specific areas of the brain through the scalp. This technique can be used to target several pathophysiological aspects in patients with AD, including the blood–brain barrier and glymphatic flow. It is also monitored by EEG [292]. Enhanced clearance of Aβ molecules is one of the main proposed mechanisms of this therapeutic approach, which is why it fits so well with sleep [290,291]. Early studies suggest that photobiomodulation can improve memory function and sleep quality in people with cognitive impairment [273,290,291]. Despite these promising results, the approach is still in its infancy and faces difficulties associated with light passing through the thick bone of the skull [293,294,295]. By contrast, exposure to red light through much thinner closed eyelids also stimulates the cerebral cortex, albeit via sensory pathways. This can be achieved while synchronizing oscillations at a frequency of 40 Hz, which may also promote glymphatic clearance [95,295,296]. At a frequency of 40 Hz, auditory evoked potentials were recorded during deep sleep stages [297,298]. Temporal lobe stimulation at 40 Hz prior to sleep has been shown to improve sleep quality in patients with insomnia. Taken together, the above studies clearly indicate the usefulness of using 40 Hz red light during sleep [299,300].

### 6.5. Cognitive Task During Stimulation

It was found that including a cognitive task involving attention and counting in both visual and auditory γ-stimulation sessions (sound anomaly detection) has a positive effect on the strength and duration of the γ-rhythm, promoting the spread of γ-oscillations to additional brain areas such as the frontal lobes and the hippocampus (which are not involved when participants are not required to perform a cognitive task) [123,124]. The latter is of particular interest since the hippocampus is considered a key target for AD treatment. This distinguishes γ-activity from activity in the β-range, which decreases with an increase in cognitive load [301]. The cognitive load increases both the power and coherence of γ-oscillations in the left frontotemporal and prefrontal sensor ranges during a syllable memorization task. The cortico-cortical connection between these structures increases [127]. Patterns of synchronization and cortico-cortical interactions were identified during the task of identifying sound inconsistencies, as well as visual and complex audio-visual inconsistencies [125]. Furthermore, the topography of gamma activity in the magnetoencephalogram during cognitive load largely coincided with the results of brain hemodynamics studies [126]. This indicates the significant benefit of cognitive tasks involving external γ-stimulation. Above, we examined the acetylcholine mechanism by which cognitive tasks influence γ-stimulation in more detail.

### 6.6. Motor (Behavioral) Activity During Stimulation

Recent studies in mice have shown that physical exercise combined with γ-therapy for four weeks reduces levels of amyloid and tau proteins to a greater extent than visual stimulation or exercise alone. This improves spatial learning, working memory and long-term memory. Furthermore, combining physical exercise with flickering gamma wave light improves calcium homeostasis, reduces reactive oxygen species levels and enhances cognitive abilities, mitochondrial function and neuroplasticity in mice [126,302]. Therefore, combining γ-stimulation with physical activity may be an effective way to enhance the desired effects.

### 6.7. Sound Wave Characteristics During Auditory Stimulation

As mentioned previously, a key issue is that prolonged and frequent image flickering can be hazardous to health, particularly for the eyes. In addition to the potential health risks, light flickering at 40 Hz can cause artifacts. Interestingly, human EEG results showed that the neural response evoked by synchronization with a single 40 Hz sound stimulus is weaker than the response to a visual stimulus [192]. This suggests that artifacts resulting from eye movements, such as regular blinking, are difficult to eliminate during the experiment, meaning that the associated EEG signals may be stronger than the brain signals of interest. Furthermore, keeping the eyes open may result in increased alpha activity in the occipital lobe, as well as changes in the topology and activity of different frequency bands compared to when the eyes are closed [303]. A possible solution that would make participants more comfortable and reduce artifacts would be to ask them to close their eyes. This is feasible when visual perception is not required. Eliminating medical and technical limitations could boost the popularity of auditory stimulation as a preventive measure and treatment for pathological changes in the brain, particularly given the evidence of successful γ-frequency oscillation targeting and the non-invasive nature of this method.

The effects of several new experimental conditions (sine or square wave sounds; eyes open and eyes closed conditions) using auditory stimulation to determine which condition could elicit a stronger neural response to a frequency of 40 Hz is well discribed [304]. The power of the 40 Hz signal was highest in the frontal lobes of the brain and when the eyes were closed during playback of a 40 Hz sine wave sound, compared to the other experimental conditions. Additionally, suppression of the α rhythm was observed. Furthermore, stronger suppression of the α-rhythms was observed in the prefrontal region when square signals of 40 Hz were used.

The neural response to a 40 Hz frequency evoked by a sine wave is stronger than that evoked by a square wave. This is probably because a sine wave is a simpler oscillation than a square wave, which contains more oscillatory components. The brain may find it easier to synchronize with the properties of a sine wave.

Results obtained using auditory stimulation at 40 Hz revealed increased neural activity in the parietal and prefrontal regions, where the default mode network (DMN) is located [193,194]. This suggests that the neural response during 40 Hz synchronization may be related to higher-level brain functions, such as learning, attention or memory.

### 6.8. The Influence of Glucose Metabolism on the γ-Rhythm

One manifestation of Alzheimer’s disease (AD) is disturbances in glucose metabolism, which manifest as increased oxidative stress before amyloid plaques appear [305,306]. Glucose levels are directly related to gamma oscillations. As maintaining γ-rhythms is an energy-intensive process that depends on a high rate of oxidative phosphorylation in neuronal mitochondria, which is limited by glucose availability, the authors hypothesize that insulin may be involved in the metabolic control of γ-oscillations [230]. PV+ interneurons selectively express the insulin-dependent glucose transporter GLUT4, which can provide an additional influx of glucose under conditions close to those that are limiting, such as during high-frequency γ-oscillations. Therefore, the effect of gamma stimulation may depend on glucose metabolism and the level of insulin resistance. To achieve a high-quality effect, attention must be paid to the presence of metabolic disorders in patients.

## 7. Clinical Trials

The large amount of data on the effectiveness of γ-stimulation has undoubtedly attracted the interest of the medical community. One consequence of this interest is attempts to create new medical devices and conduct clinical trials.

As of November 2025, GENUS devices have most frequently been tested or are planned to be tested in the treatment of Alzheimer’s disease and Parkinson’s disease, according to the database https://clinicaltrials.gov/ (accession date 17 November 2025).

A phase I/II randomized, controlled, single-blind multi-center clinical trial of the GammaSense Stimulation device medical device is currently underway. This device is intended for use in the treatment of prodromal AD, sever AD or Mild Cognitive Impairment (MCI) in men and women over the age of 55 years. During the trials, participants underwent daily gamma stimulation for 1 h/day for 6 months. The results of the study have already been obtained and published, according to which gamma stimulation partially relieves the symptoms of AD, improves the results of the Mental State Examination and ADSC-ADL (Instrumental, basic, and total) tests, and improves sleep quality [63,307]. However, in addition to the therapeutic effects, side effects of γ-stimulation were also found. The most common adverse effects were ear and labyrinth disorders, tinnitus, and headache (15–21%). Musculoskeletal and connective tissue, skin and subcutaneous tissue disorders occurred in 8–13% of participants, which is less frequent but higher than in the placebo group [307]. Clinical trial data indicate the need to modify the stimulation protocol to minimize the undesirable side effects of gamma stimulation.

A clinical trial of the GENUS device (NCT05655195) is also planned for individuals of both sexes diagnosed with AD aged 65–100 years. Participants will undergo daily combined (light and sound) stimulation at 40 Hz for 1 h per day, every day for 6 months, followed by analysis of changes in functional brain connectivity, cognitive functions, sleep/wake patterns and molecular biomarkers of AD. In addition to potential therapeutic effects, potential side effects (headache, changes in emotional status, etc.) and other risks associated with gamma stimulation will be evaluated.

In addition to AD, participants are being recruited for clinical trials of the use of 40 Hz frequency light, sound, and tactile stimulation in the treatment of Parkinson’s disease (restoration of motor functions) NCT05268887. Clinical trials will be conducted on individuals aged 45–90 with a confirmed diagnosis. The study is based on the results of randomized trials in patients with Parkinson’s disease undergoing deep brain stimulation at frequencies of 130–140 Hz [308,309,310]. However, in this case, it is difficult to speak of classical γ-stimulation, since both the central frequency (133–140 Hz as opposed to 40 Hz) and the method of stimulation (direct electrical stimulation of deep brain centers using an implantable device) differ significantly. In this case, there are significant risks of side effects from invasive intervention, and the highest qualifications of the operating neurosurgeon are required. Therefore, the planned medical trial will evaluate the efficacy and safety of γ-stimulation in the treatment of Parkinson’s disease.

The initial success of gamma stimulation as a potential therapeutic approach for AD and Parkinson’s disease [62] suggests its possible effectiveness in the treatment of other disorders. In particular, participants of both sexes aged 18–75 with depressive disorders are being recruited to test the therapeutic potential and safety of GENUS (NCT05680220). Unfortunately, the publicly available documentation does not allow us to evaluate all the characteristics of the stimulating effect: the wavelength of the stimulating light, the frequency and volume of the sound, and the characteristics of the tactile stimuli (if any). Therefore, the use of these clinical studies at this stage of our meta-analysis is difficult. We are confident that upon completion of the clinical studies, their results will be published and available to supplement the overall picture.

## 8. Limitations and Prospects

We attempted to systematize the literature on the effects of γ-stimulation and identify the relationship between its effectiveness and various factors such as the nature and frequency of the stimulus, the species and age of the organism, and the presence of pathology. We confirmed and expanded upon existing data regarding changes in the range of effective stimulation frequencies with age. We also identified differences in interspecies responses to γ-stimulation. We hope that our results will help future researchers to design more effective and efficient experiments on this topic.

However, like any other work in this area, this work has a number of limitations. Firstly, not all characteristics of periodic stimuli could be analyzed. In particular, there are no unified units of measurement for LED luminance in the case of a visual signal. Different studies use different characteristics and units of measurement for light sources, such as luminous flux (lm), illuminance (lx), luminous intensity (cd), luminous intensity per area (cd/cm^2^) or power (mW/cm^2^). Some units can be relatively accurately converted to each other (e.g., lx and lm, or lx and cd/cm^2^), if the setup geometry is described in detail. However, the data is often insufficient for accurate calculations [39,54,96,311]. Some pairs of units cannot be converted to each other in principle based on published data (e.g., mW/cm^2^ and cd/cm^2^). Furthermore, even if we could accurately estimate the luminous flux, luminance or power of a light source, we would still be unable to estimate the proportion of light reaching the eye in each specific case. If necessary, a more in-depth study could be conducted using additional sources, standardized units of measurement for light source characteristics, and other exposure conditions. However, this was beyond the scope of this review.

In the case of sound stimulation, data is often given either on the sound level (dB), but frequency or power stay unclear [303].

Secondly, several interesting characteristics and methodological requirements were excluded from the analysis due to insufficient data. These included whether the eyes were open or closed, the magnitude of the contrast (Michelson contrast), the change in the angle of inclination of the image when the visual stimulus was presented, and the presence of a tactile stimulus [94,95,117,138]. We admit that each of the presented additional conditions may be important for the realization of the effects of γ-stimulation. However, statistical verification of this requires either an expansion of the analyzed literature base or additional experimental work in this area.

Thirdly, we employed relatively straightforward and speedy methods to evaluate the quantitative relationships between the effectiveness of γ-stimulation and its frequency, type, duration and the age of the subjects. The use of more sensitive multivariate approaches, such as principal component analysis and multivariate correlation, as well as AI for the automated analysis of primary data, may allow us to identify additional patterns and refine existing relationships.

For future research, it is important to clarify the relationship between the effectiveness of γ-stimulation in young (under 20 years) and middle-aged (30–50 years) individuals, as there are clearly gaps in this area. As a significant proportion of research into the potential therapeutic applications of γ-stimulation is conducted in mice, it is necessary to search for new, more valid models, particularly for Alzheimer’s disease (AD). Furthermore, the reproducibility of results between mice and humans must be carefully compared. In human studies, it is crucial that electrophysiological results are validated by behavioral and/or cognitive tests. We hope that the proportion of such comprehensive studies will increase in the future.

Medical history and the use of certain medications can have a significant impact on the effects of γ-stimulation in humans. We believe that analyzing large samples of patients and healthy volunteers, taking their medical histories into account, will significantly improve our understanding of how stimulus characteristics affect γ-stimulation.

## 9. Conclusions

The γ-oscillations play a crucial role in how the brain perceives, processes and stores information. Stimulation with synchronized light and sound (or other methods) at a frequency of 40 Hz effectively induces corresponding brain activity at the same frequency. Overall, stimulation of the brain at 40 Hz is associated with reduced neuroinflammation, enhanced synaptic transmission, and increased expression of genes associated with synaptic plasticity. PV+ and SST+ interneurons are sensitive to this stimulation. This results in increased coherence and activation of neural networks, particularly DMN, executive network and salience network, leading to improved cognitive function. Exposure to 40 Hz also increases cytokine production and phagocytic activity in microglia, normalizes circadian rhythms and increases the rate of glymphatic clearance of Aβ. Collectively, these effects lead to a reduction in amyloid plaque formation.

Human studies do not always yield definitive results. There is reason to believe that γ-stimulation requires careful selection of parameters, taking into account the central γ-frequency, hue, brightness and exposure duration. This automatically takes into account age-related changes and other physiological characteristics of the individual, such as the level of GABA production, as well as the possible contribution of selective insulin resistance of PV+ interneurons to the progression of CNS diseases associated with cognitive impairment.

To enhance the effects of γ-stimulation, such as broader coverage and stimulation of deep brain regions, cognitive tasks, physical activity and a combination of visual and auditory stimulation must be used. Furthermore, gentler stimulation methods must be found to avoid unwanted reactions to strobe light. The approach of stimulation during sleep is very interesting.

Special attention should be paid to the use of medications during γ-therapy. Some substances can reduce the effect of gamma synchronization, in particular the calcium channel blocker felodipine. Conversely, NMDA receptor antagonists (of which there are many drugs) can enhance the power of abnormal γ-oscillations. The expression of chloride and sodium ion channels, and therefore the substances that block them, also influences γ-synchronization. Furthermore, glucose metabolism and insulin resistance should be considered, as these can affect the outcome of γ-stimulation.

The γ-stimulation effects have been found to vary depending on species, age, frequency and stimulus type. Visual stimulation protocols using white light are the most effective in humans. The range of effective central frequencies narrows from 20–60 Hz to 30–40 Hz as people age. While some studies have observed a decrease in the central frequency, others have shown an increase above 40 Hz for individuals aged 30–50. These findings must be taken into account when selecting an individual stimulation regimen.

AD patients and mouse models respond differently to γ-stimulation, so special attention should be paid to the selection of study subjects when assessing therapeutic potential. Additionally, cognitive and behavioral responses are less sensitive to gamma stimulation than biochemical and EEG responses. Therefore, it is advisable to supplement EEG data with behavioral studies to avoid false-positive results.

Our results may be useful in deepening the understanding of the mechanisms of gamma stimulation, when planning future experiments and in the search for potential therapeutic regimens of γ-stimulation.

## Figures and Tables

**Figure 1 biology-14-01722-f001:**
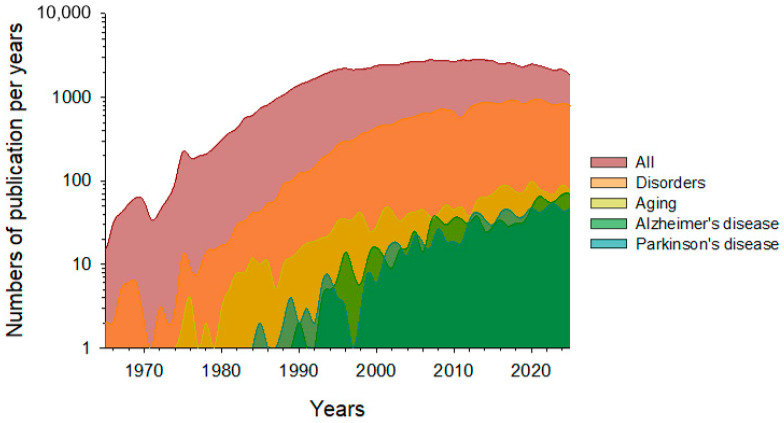
The dynamics of global publication activity on specific topics in gamma stimulation research. The search results for the following keywords are presented: ‘gamma stimulation’ (maroon line and fill), ‘gamma stimulation’ + ‘disorders’ (orange line and fill), ‘gamma stimulation’ + ‘aging’ (yellow line and fill), ‘gamma stimulation’ + ‘Alzheimer’s disease’ (green line and fill) and ‘gamma stimulation’ + ‘Parkinson’s disease’ (blue line and fill). The data were extracted from PubMed (https://pubmed.ncbi.nlm.nih.gov; accession date: 22 October 2025).

**Figure 2 biology-14-01722-f002:**
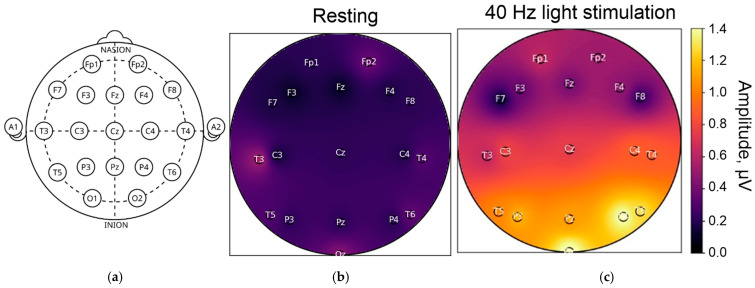
Brain response to 40 Hz stimulation during visual stimulation, according to our own experimental data: diagram of the electrode placement during EEG-recording (**a**), EEG in the frequency range of 39–41 Hz in resting state without stimulation (**b**) and brain response on 40 Hz visual stimulation (**c**). Resting and response amplitudes were recorded in equal conditions and shown in μV.

**Figure 3 biology-14-01722-f003:**
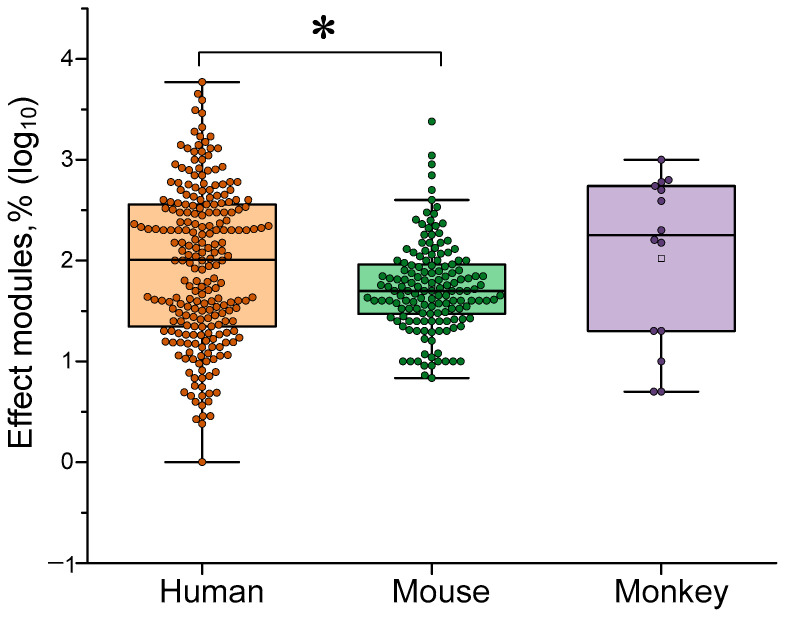
A comparison of the efficacy of gamma stimulation in different species: humans, mice and marmosets. The data are presented as median values and percentiles of 10%, 25%, 75%, and 90%. Each point corresponds to an experimental value taken from the literature. Each article may contain at least one experimental point. A total of 408 experimental values from 40 published papers were analyzed. *—*p* < 0.05, Kruskal–Wallis ANOVA with post hoc Dunn’s test.

**Figure 4 biology-14-01722-f004:**
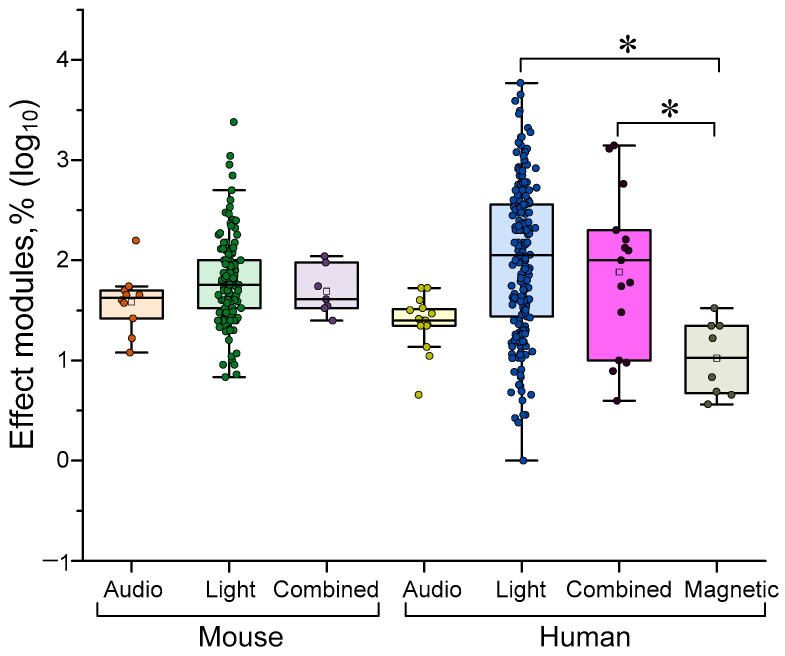
A comparison of the efficacy of gamma-stimulation methods for different species. Data are presented as median values and percentiles at the 10th, 25th, 75th and 90th percentiles. Each point corresponds to an experimental value taken from the literature. Each article may contain at least one experimental point. A total of 393 experimental values from 38 published studies were analyzed. *—*p* < 0.05, Kruskal–Wallis ANOVA with post hoc Dunn’s test.

**Figure 5 biology-14-01722-f005:**
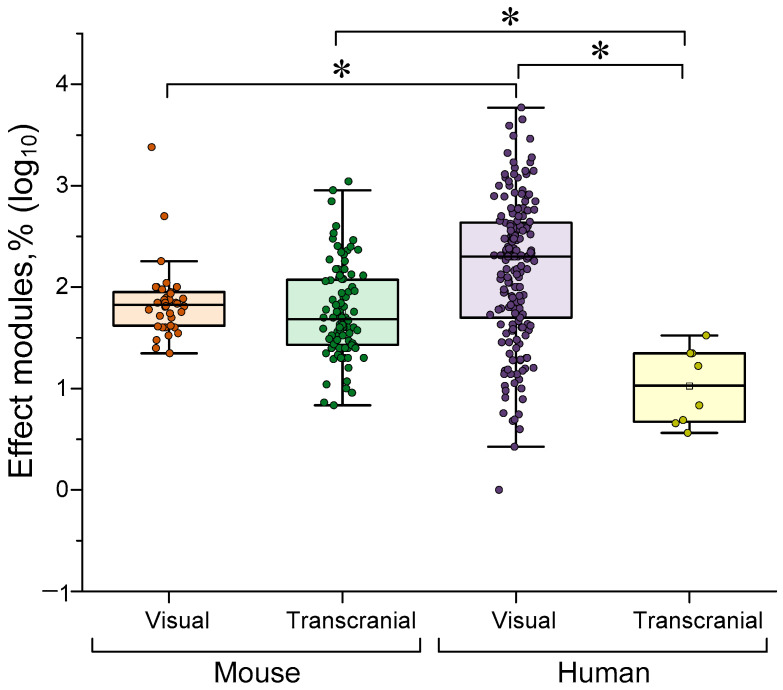
A comparison of the efficacy of different methods of exposure to light γ-stimulation. The data are presented as median values and percentiles of 10%, 25%, 75% and 90%. Each point corresponds to an experimental value taken from the literature. Each article may contain at least one experimental point. A total of 323 experimental values from 38 published studies were analyzed. *—*p* < 0.05, Kruskal–Wallis ANOVA with post hoc Dunn’s test.

**Figure 6 biology-14-01722-f006:**
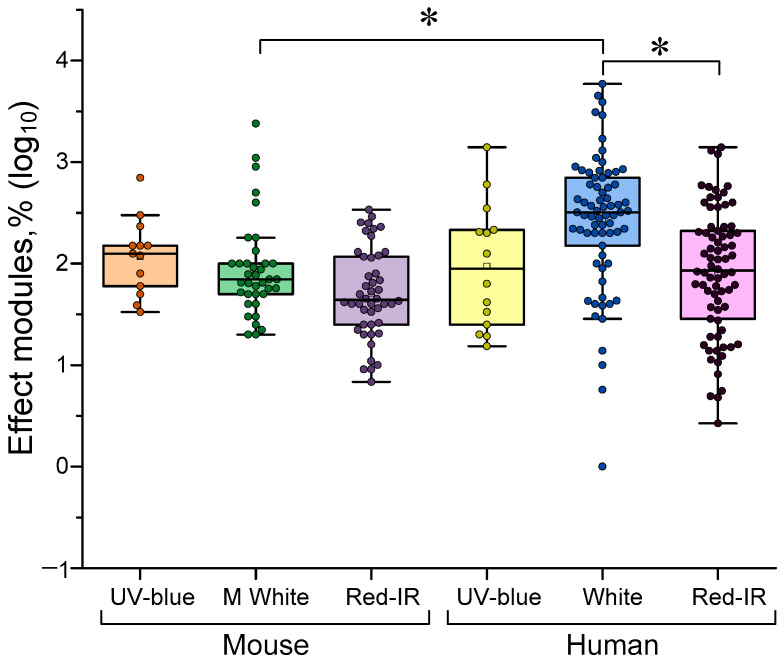
The efficiency of light γ-stimulation depending on the wavelength range used: 360–470 nm (UV–blue), 500–600 nm (white) and ≥600 nm (red–IR). The data are presented as median values and percentiles of 10%, 25%, 75% and 90%. Each point corresponds to an experimental value taken from the literature. At least one experimental point can be taken from each article. A total of 323 experimental values from 38 published studies were analyzed. *—*p* <0.05, Kruskal–Wallis ANOVA with post hoc Dunn’s test.

**Figure 7 biology-14-01722-f007:**
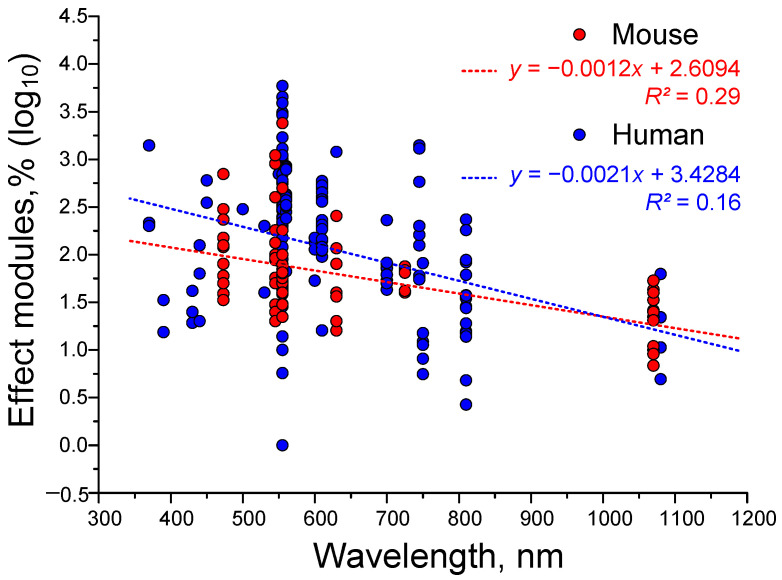
The dependence of the efficiency of light γ-stimulation on wavelength in humans (blue dots) and mice (red dots). Each dot represents an experimental value taken from the literature. Each article may contain at least one data point. The dotted lines correspond to straight lines of approximation for humans (blue) and mice (red). A total of 323 experimental values from 38 published papers were analyzed.

**Figure 8 biology-14-01722-f008:**
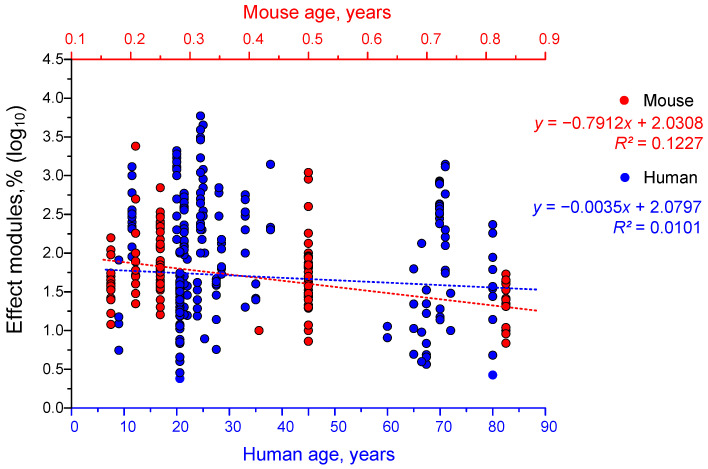
The dependence of the effectiveness of γ-frequency light stimulation on age in humans (blue dots) and mice (red dots) in years. Each dot corresponds to one experimental value taken from the literature. Each article may contain at least one experimental data point. The dotted lines correspond to the straight lines of approximation for humans (blue) and mice (red). A total of 393 experimental values from 38 published papers were analyzed.

**Figure 9 biology-14-01722-f009:**
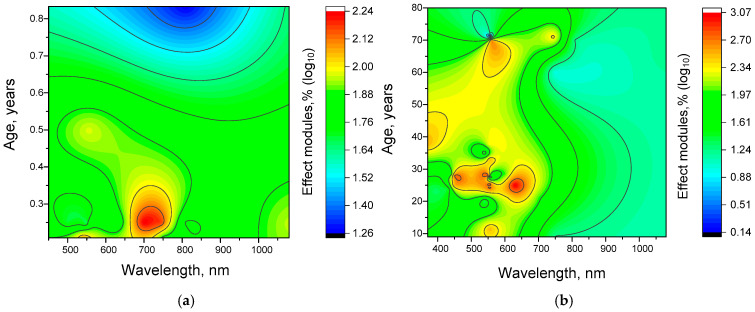
Three-dimensional maps of the dependence of the effectiveness of gamma stimulation effects on the wavelength of the stimulation light (abscissa) and age (ordinate). Reconstructions were constructed using the Kriging correlation method based on the points shown in Figure 4 and Figure 5. (**a**) Mouse, (**b**) Human.

**Figure 10 biology-14-01722-f010:**
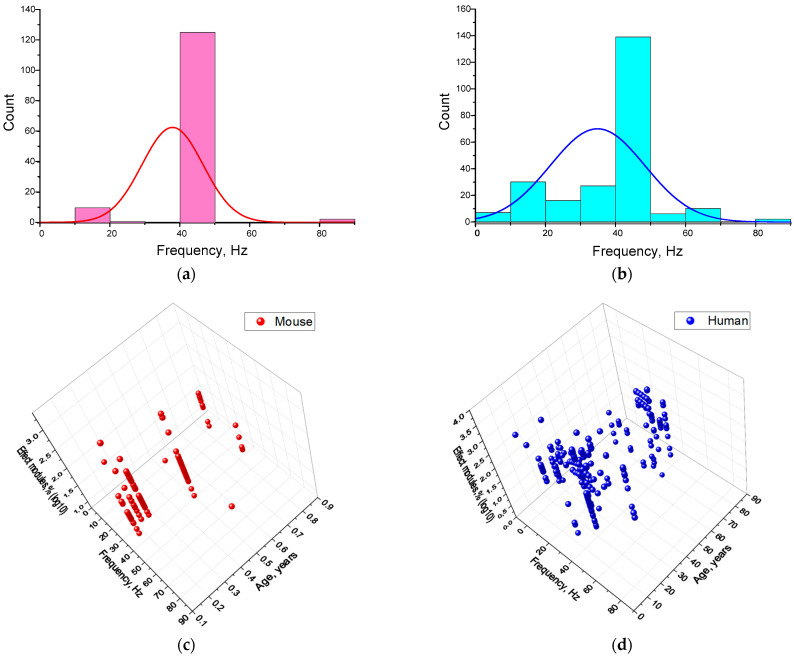
The efficiency of γ-stimulation effects depends on the frequency of stimulation. Frequency distributions used for γ-stimulation in mice (**a**) and humans (**b**). The bar graphs show the actual, discrete distribution of frequencies used in the analyzed studies. The lines represent the fitted theoretical normal distribution for each diagram. Three-dimensional scatter plots (**c**,**d**) and 3D maps (**e**,**f**) showing the dependence of the γ-stimulation effect modules (*z*-axis) on frequency (abscissae) and age (ordinate) for mice (**c**,**e**) and humans (**d**,**f**). Each point in graphs (**c**,**e**) corresponds to one experimental value from the analyzed studies. Each article may contain at least one experimental point. Three-dimensional reconstructions were constructed using the Kriging correlation method based on the points. A total of 408 experimental values from 40 published studies were analyzed (144 points for mice and 249 for humans).

**Figure 11 biology-14-01722-f011:**
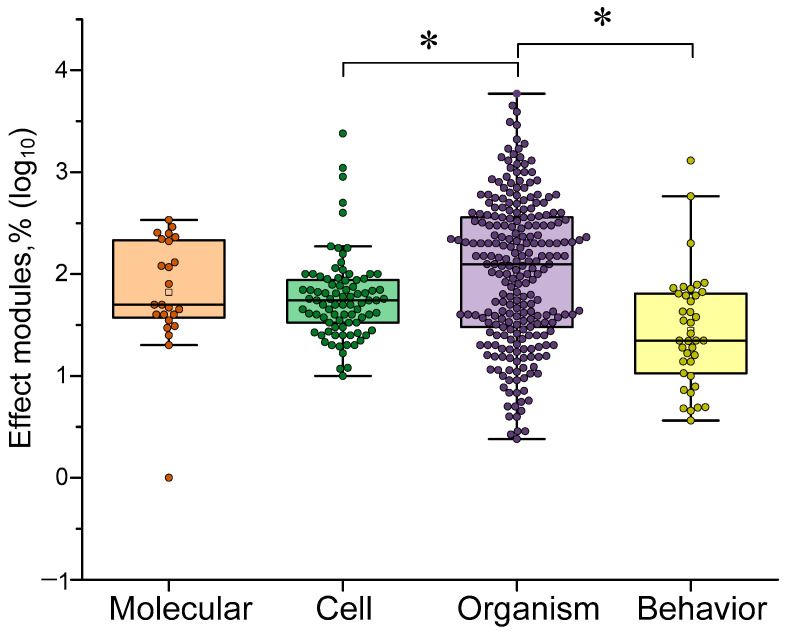
The comparison of sensitivity of different aspects (molecular, cellular, organismal and behavior) of γ-stimulation. The data are presented as median values and percentiles of 10%, 25%, 75%, and 90%. Each point corresponds to an experimental value taken from the literature. Each article may contain at least one experimental point. A total of 408 experimental values from 40 published papers were analyzed. *—*p* < 0.05, Kruskal–Wallis ANOVA with post hoc Dunn’s test.

**Figure 12 biology-14-01722-f012:**
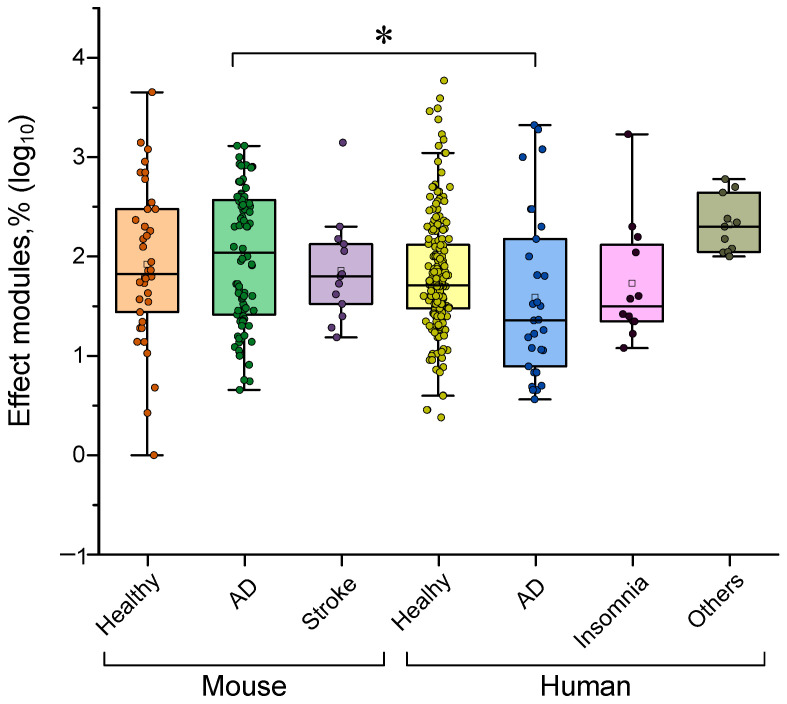
γ-stimulation efficiency in cases involving different species and the pathology under study. The data are presented as median values and percentiles of 10%, 25%, 75% and 90%. Each point corresponds to an experimental value taken from the literature. Each article may contain at least one experimental point. A total of 393 experimental values from 38 published studies were analyzed. *—*p* < 0.05, Kruskal–Wallis ANOVA with post hoc Dunn’s test.

**Figure 13 biology-14-01722-f013:**
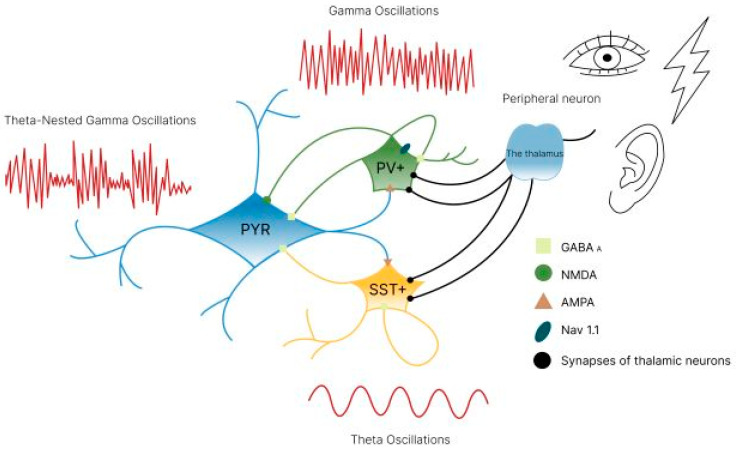
The mechanism of γ-oscillations. The functional connection between pyramidal neurons (PYR) and inhibitory interneurons expressing parvalbumin (PV+) and somatostatin (SST+) dynamically organizes synchronous oscillations in the θ- and γ-ranges via γ-oscillation mechanisms in the pyramidal neuron-interneuron network (PING) or in the interneuron network (ING) (references in the text).

**Figure 14 biology-14-01722-f014:**
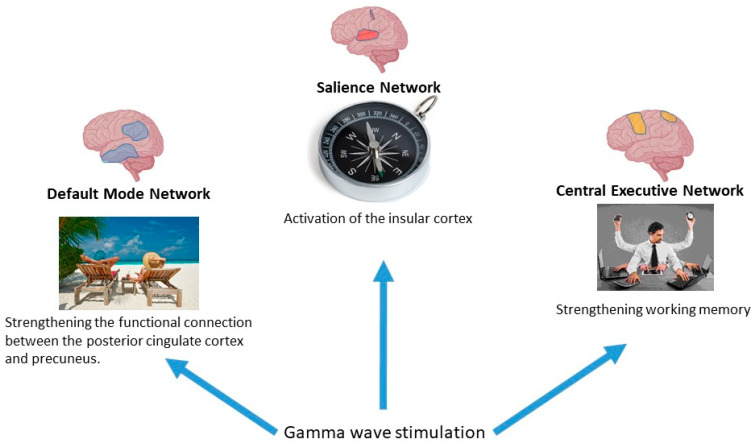
The most studied neural networks of the brain.

## Data Availability

The data presented in this study are available in Appendix A, Table A1.

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
