# Peer review of "Brain Gamma-Stimulation: Mechanisms and Optimization of Impact"

_biology, 2025, doi:10.3390/biology14121722_

Round 1

Reviewer 1 Report

Comments and Suggestions for Authors

This review is a large-scale work dedicated to analyzing the efficacy of non-invasive γ-stimulation (at a frequency of ~30-60 Hz) for modulating brain activity and its potential therapeutic applications, primarily for neurodegenerative diseases such as Alzheimer's disease. The primary merit of the review is its attempt to transition from a qualitative description to a quantitative analysis. The authors do not merely summarize the literature data but conduct a meta-analysis, converting heterogeneous results into a unified logarithmic scale. This allows for the comparison of stimulation efficacy across different studies.

A few minor remarks:

  1. Study selection criteria: For such a comprehensive literature review, it is necessary to clearly specify the criteria used for including articles in the analysis (databases, keywords, time frame).

  2. Add references and a section discussing clinical trials: It would be beneficial to add a section discussing clinical studies that are completed or currently ongoing (e.g., https://www.clinicaltrials.gov/study/NCT05655195, https://clinicaltrials.gov/study/NCT03556280, etc.).

  3. Line 167-169: The sentence "In 2009, significant data were obtained on the importance of rapidly excitable PV+ interneurons for the emergence and maintenance of γ-oscillations, primarily in the 40 Hz range [52,53]" is supported by references [52,53] from 1998 and 2001, respectively. This should be clarified or corrected.

The presented scientific review is an outstanding work that makes a significant contribution to the field of neuroscience. Its greatest strengths are the systematic quantitative analysis, the identification of complex "stimulation parameter-efficacy" relationships, and the in-depth elaboration of molecular-cellular mechanisms. Despite some remarks, which primarily concern methodological details and structure, the review is highly valuable both for researchers planning experiments in this field and for clinicians considering γ-stimulation as a potential therapeutic tool. It is recommended for publication after minor revisions.

Author Response

This review is a large-scale work dedicated to analyzing the efficacy of non-invasive γ-stimulation (at a frequency of ~30-60 Hz) for modulating brain activity and its potential therapeutic applications, primarily for neurodegenerative diseases such as Alzheimer's disease. The primary merit of the review is its attempt to transition from a qualitative description to a quantitative analysis. The authors do not merely summarize the literature data but conduct a meta-analysis, converting heterogeneous results into a unified logarithmic scale. This allows for the comparison of stimulation efficacy across different studies.

We thank Reviewer for carefully reviewing the manuscript and providing valuable recommendations. Responses to comments are highlighted in red.

A few minor remarks:

  1. Study selection criteria:For such a comprehensive literature review, it is necessary to clearly specify the criteria used for including articles in the analysis (databases, keywords, time frame).

Reply 1:

We thank the reviewer for their valuable input.

A literature search was performed using the publicly available databases PubMed (https://pubmed.ncbi.nlm.nih.gov/), Google Scholar (https://scholar.google.com/), and the websites of major publishers: Nature (https://www.nature.com/), MDPI (https://www.mdpi.com/), and others. The search was performed using the keywords 'gamma stimulation', 'gamma frequency stimulation', or '50 Hz brain rhythm stimulation'. To refine the search, we used combinations with additional keywords, such as: 'gamma stimulation' + 'disorders' (Alzheimer's disease, Parkinson's disease, etc.), 'gamma stimulation' + 'aging', 'gamma stimulation' + 'sleeping', 'gamma stimulation' + 'sound' or 'light', 'gamma stimulation' + 'human' or 'animals', etc. When including articles in the analysis, preference was given to articles published in the last 10 years (2015-2025) and 20 years (2005-2025). The shares of these works were 62% and 89%, respectively. Works published earlier (before 2005) were included if they contained pioneering data, important results for understanding the mechanisms of gamma rhythm functioning, or results that were not published in later periods. Publications that included completely identical experimental conditions to those already added, or those with insufficiently well-described experimental conditions (duration, frequency, nature of the stimulus, and its characteristics) were excluded from the analysis.

This addition was made to the manuscript and highlighted in a separate section, "Criteria for Including Publications in the Analysis."

  1. Add references and a section discussing clinical trials:It would be beneficial to add a section discussing clinical studies that are completed or currently ongoing (e.g., https://www.clinicaltrials.gov/study/NCT05655195, https://clinicaltrials.gov/study/NCT03556280, etc.).

Reply 2:

The large amount of data on the effectiveness of γ-stimulation has undoubtedly attracted the interest of the medical community. One consequence of this interest is attempts to create new medical devices and conduct clinical trials.

As of November 2025, GENUS devices have most frequently been tested or are planned to be tested in the treatment of Alzheimer's disease and Parkinson's disease, according to the database https://clinicaltrials.gov/ (accession date 17.11.2025).

A phase I/II randomised, controlled, single-blind multi-centre clinical trial of the GammaSense Stimulation device medical device is currently underway. This device is intended for use in the treatment of prodromal AD, sever AD or Mild Cognitive Impairment (MCI) in men and women over the age of 55 years. During the trials, participants underwent daily gamma stimulation for 1 hour/day for 6 months. The results of the study have already been obtained and published, according to which gamma stimulation partially relieves the symptoms of AD, improves the results of the Mental State Examination and ADSC-ADL (Instrumental, basic, and total) tests, and improves sleep quality [1,2]. However, in addition to the therapeutic effects, side effects of γ-stimulation were also found. The most common adverse effects were ear and labyrinth disorders, tinnitus, and headache (15-21%). Musculoskeletal and connective tissue, skin and subcutaneous tissue disorders occurred in 8-13% of participants, which is less frequent but higher than in the placebo group [2]. Clinical trial data indicate the need to modify the stimulation protocol to minimise the undesirable side effects of gamma stimulation.

A clinical trial of the GENUS device (NCT05655195) is also planned for individuals of both sexes diagnosed with AD aged 65-100 years. Participants will undergo daily combined (light and sound) stimulation at 40 Hz for 1 hour per day, every day for 6 months, followed by analysis of changes in functional brain connectivity, cognitive functions, sleep/wake patterns and molecular biomarkers of AD. In addition to potential therapeutic effects, potential side effects (headache, changes in emotional status, etc.) and other risks associated with gamma stimulation will be evaluated.

In addition to AD, participants are being recruited for clinical trials of the use of 40 Hz frequency light, sound, and tactile stimulation in the treatment of Parkinson's disease (restoration of motor functions) NCT05268887. Clinical trials will be conducted on individuals aged 45-90 with a confirmed diagnosis. The study is based on the results of randomised trials in patients with Parkinson's disease undergoing deep brain stimulation at frequencies of 130-140 Hz [3-5]. However, in this case, it is difficult to speak of classical γ-stimulation, since both the central frequency (133-140 Hz as opposed to 40 Hz) and the method of stimulation (direct electrical stimulation of deep brain centres using an implantable device) differ significantly. In this case, there are significant risks of side effects from invasive intervention, and the highest qualifications of the operating neurosurgeon are required. Therefore, the planned medical trial will evaluate the efficacy and safety of γ-stimulation in the treatment of Parkinson's disease.

The initial success of gamma stimulation as a potential therapeutic approach for AD and Parkinson's disease [6] suggests its possible effectiveness in the treatment of other disorders. In particular, participants of both sexes aged 18-75 with depressive disorders are being recruited to test the therapeutic potential and safety of GENUS (NCT05680220). Unfortunately, the publicly available documentation does not allow us to evaluate all the characteristics of the stimulating effect: the wavelength of the stimulating light, the frequency and volume of the sound, and the characteristics of the tactile stimuli (if any). Therefore, the use of these clinical studies at this stage of our meta-analysis is difficult. We are confident that upon completion of the clinical studies, their results will be published and available to supplement the overall picture.

We add all this information in the separate section.

  1. Line 167-169:The sentence "In 2009, significant data were obtained on the importance of rapidly excitable PV+ interneurons for the emergence and maintenance of γ-oscillations, primarily in the 40 Hz range [52,53]" is supported by references [52,53] from 1998 and 2001, respectively. This should be clarified or corrected.

Reply 3:

We thanks Reviewer for importance addition and supplement this part:

In 2009, significant data were obtained on the importance of rapidly excitable PV+ interneurons for the emergence and maintenance of γ-oscillations, primarily in the 40 Hz range [52,53]. Later, the role of neuronal plasticity of CP-AMPARs- and mGluRs-dependent calcium signaling of PV+ interneurons and their interaction with Sst+ interneurons in maintaining γ-oscillations was clarified [7-9].

The presented scientific review is an outstanding work that makes a significant contribution to the field of neuroscience. Its greatest strengths are the systematic quantitative analysis, the identification of complex "stimulation parameter-efficacy" relationships, and the in-depth elaboration of molecular-cellular mechanisms. Despite some remarks, which primarily concern methodological details and structure, the review is highly valuable both for researchers planning experiments in this field and for clinicians considering γ-stimulation as a potential therapeutic tool. It is recommended for publication after minor revisions.

References

  1. Cimenser, A.; Hempel, E.; Travers, T.; Strozewski, N.; Martin, K.; Malchano, Z.; Hajós, M. Sensory-Evoked 40-Hz Gamma Oscillation Improves Sleep and Daily Living Activities in Alzheimer’s Disease Patients. Frontiers in Systems Neuroscience 2021, 15, doi:10.3389/fnsys.2021.746859.
  2. Hajós, M.; Boasso, A.; Hempel, E.; Shpokayte, M.; Konisky, A.; Seshagiri, C.V.; Fomenko, V.; Kwan, K.; Nicodemus-Johnson, J.; Hendrix, S.; et al. Safety, tolerability, and efficacy estimate of evoked gamma oscillation in mild to moderate Alzheimer’s disease. Frontiers in Neurology 2024, 15, doi:10.3389/fneur.2024.1343588.
  3. Deuschl, G.; Schade-Brittinger, C.; Krack, P.; Volkmann, J.; Schäfer, H.; Bötzel, K.; Daniels, C.; Deutschländer, A.; Dillmann, U.; Eisner, W.; et al. A Randomized Trial of Deep-Brain Stimulation for Parkinson's Disease. New England Journal of Medicine 2006, 355, 896-908, doi:10.1056/NEJMoa060281.
  4. Muthuraman, M.; Bange, M.; Koirala, N.; Ciolac, D.; Pintea, B.; Glaser, M.; Tinkhauser, G.; Brown, P.; Deuschl, G.; Groppa, S. Cross-frequency coupling between gamma oscillations and deep brain stimulation frequency in Parkinson’s disease. Brain 2020, 143, 3393-3407, doi:10.1093/brain/awaa297.
  5. Okun, M.S.; Fernandez, H.H.; Wu, S.S.; Kirsch‐Darrow, L.; Bowers, D.; Bova, F.; Suelter, M.; Jacobson, C.E.; Wang, X.; Gordon, C.W.; et al. Cognition and mood in Parkinson's disease in subthalamic nucleus versus globus pallidus interna deep brain stimulation: The COMPARE Trial. Annals of Neurology 2009, 65, 586-595, doi:10.1002/ana.21596.
  6. Adaikkan, C.; Middleton, S.J.; Marco, A.; Pao, P.-C.; Mathys, H.; Kim, D.N.-W.; Gao, F.; Young, J.Z.; Suk, H.-J.; Boyden, E.S.; et al. Gamma Entrainment Binds Higher-Order Brain Regions and Offers Neuroprotection. Neuron 2019, 102, 929-943.e928, doi:10.1016/j.neuron.2019.04.011.
  7. Hadler, M.D.; Tzilivaki, A.; Schmitz, D.; Alle, H.; Geiger, J.R.P. Gamma oscillation plasticity is mediated via parvalbumin interneurons. Science Advances 2024, 10, doi:10.1126/sciadv.adj7427.
  8. Onorato, I.; Tzanou, A.; Schneider, M.; Uran, C.; Broggini, A.C.; Vinck, M. Distinct roles of PV and Sst interneurons in visually induced gamma oscillations. Cell Reports 2025, 44, doi:10.1016/j.celrep.2025.115385.
  9. Zhang, N.; Hu, B.-W.; Li, X.-M.; Huang, H. Rethinking parvalbumin: From passive marker to active modulator of hippocampal circuits. IBRO Neuroscience Reports 2025, 19, 760-773, doi:10.1016/j.ibneur.2025.10.005.

Reviewer 2 Report

Comments and Suggestions for Authors

Lushnikov et al have provided an extended, detailed, and very well-integrated analysis of Brain Gamma-Stimulation: Mechanisms And Optimization of Impact. The authors had performed tremendous work describing the current state of brain gamma stimulation. However, before this manuscript can be accepted, the authors have to address a list of major and minor comments

Major

  • Please create an image of the different brain areas, which will support the comprehensive understanding of different areas of the brain affected by different oscillation waves
  • Please provide a more detailed explanation of the effect modules. What this metric represents in each section
  • Please elaborate sections 4.4.2 (4.4.2. The default mode network), 4.3.2 (4.3.2. Central executive network), 4.4.3 (4.4.3. Salience network executive network). Provide more explanation, how these effects are presented and what consequences for the health benefits it can have
  • Please add the clinical translation and protocols section, which will include a set of tables that will summarize each effect, each wavelength, and studies supporting it as well a possible negative effect on the health for humans and animal models.
  • Authors make a great overview of the field with multiple details, now they need to write a section, which will summarize all the beneficial effects, protocols/waves needed for it, studies supporting it and what can be most adequate plan to use all known wavehterapies for the health, longevity and quality of life
  • Please reconcile wavelength claims by explaining when red/long wave vs white/500-600 nm stimuli are superior and why

MINOR

  • Please remove ‘it is believed’ and change it to ‘it is currently suggested’ line 89
  • Please standardize the numbering orders and sections to ensure correct cross-reference matches

Author Response

Lushnikov et al have provided an extended, detailed, and very well-integrated analysis of Brain Gamma-Stimulation: Mechanisms And Optimization of Impact. The authors had performed tremendous work describing the current state of brain gamma stimulation. However, before this manuscript can be accepted, the authors have to address a list of major and minor comments

We thank Reviewer for carefully reviewing the manuscript and providing valuable recommendations. Responses to comments are highlighted in red.

Major

  1. Please create an image of the different brain areas, which will support the comprehensive understanding of different areas of the brain affected by different oscillation waves

Reply 1:

We thank Reviewer for important recommendation.

These rhythms have been recorded and studied in the cerebral cortex, the hippocampus, the amygdala, the olfactory bulb, the striatum, and the brainstem [12,19]. Figure 2 shows the results of our own experimental studies. It is clear that the parietal and occipital regions respond most actively to stimulation. In our own studies (as yet unpublished), we also tested the brain's response to visual stimulation in the range of 30 to 50 Hz. The response is highly individual. Therefore, the patterns of response of different brain regions to gamma stimulation at different frequencies remain to be fully described and elucidated.

(Please, see attached file)

Figure 2. Brain response to 40 Hz stimulation during visual stimulation, according to our own experimental data: diagram of the electrode placement during EEG-recording (a), EEG in the frequency range of 39-41 Hz in resting state without stimulation (b) and brain response on 40 Hz visual stimulation (c). Resting and response amplitudes were recorded in equal conditions and shown in μV.

We add new figure in the text.

  1. Please provide a more detailed explanation of the effect modules. What this metric represents in each section

Reply 2:

As described previously, to enable comparison of disparate effects, we first converted to dimensionless quantities and calculated the effects of the treatments. We calculated the effects using formula (1)

                      Please, see attached file        (1),

where A is the initial value of the measured characteristic (EEG potential, proportion of differentiated cells, mental performance scores, etc.), and ΔA is the value of the same parameter after γ-stimulation (in the same units as A). Expressing the values as percentages allows us to convert to dimensionless quantities and compare the effectiveness of γ-stimulation for different targets (Figure 10). Taking the absolute values allows us to adequately average the contributions of disparate effects. Further logarithmization allowed us to adequately assess the significance of statistical differences between groups, where effect sizes vary from units to hundreds of percent.

The initial quantitative values of A and ΔA for the analysis were taken from the texts and/or graphs of the analyzed manuscripts.

This clarification has been added to the manuscript.

  1. Please elaborate sections 4.4.2 (4.4.2. The default mode network), 4.3.2 (4.3.2. Central executive network), 4.4.3 (4.4.3. Salience network executive network). Provide more explanation, how these effects are presented and what consequences for the health benefits it can have.

Peply 3:

Surprisingly, despite the significant importance of understanding that the brain functions as a network and that gamma oscillations mediate communication between different brain regions, we found no studies in the literature on the effects of gamma stimulation on the activity of known neural networks. The few literature sources we were able to find demonstrating the possibility of such an effect do not yet provide a comprehensive picture and are quite sparse. We would like to draw the attention of researchers to this extremely interesting area of research.

Here we present information on three of the most studied neural networks and our findings on their relationship with γ-stimulation.

Thus, it is clear that research is limited, and at present, only the possibility of such an effect exists. Nevertheless, we felt it was necessary to include this information in this article. Since it has been established that the functional activity of these neural networks is altered in a wide range of diseases, establishing the effect of gamma stimulation on their functioning could be very promising and open up additional possibilities for non-drug therapy.

  1. Please add the clinical translation and protocols section, which will include a set of tables that will summarize each effect, each wavelength, and studies supporting it as well a possible negative effect on the health for humans and animal models.

Reply 4:

We thank Reviewer for their valuable recommendation.

Unfortunately, a complete list of side effects of γ-stimulation is currently unavailable. Clinical trials addressing this issue are currently being planned. В частности, a clinical trial of the GENUS device (NCT05655195) is also planned for individuals of both sexes diagnosed with AD aged 65-100 years. Participants will undergo daily combined (light and sound) stimulation at 40 Hz for 1 hour per day, every day for 6 months, followed by analysis of changes in functional brain connectivity, cognitive functions, sleep/wake patterns and molecular biomarkers of AD. In addition to potential therapeutic effects, potential side effects (headache, changes in emotional status, etc.) and other risks associated with gamma stimulation will be evaluated.

Specific cases of side effects and ways to minimize them have already been discussed in the "Optimization of Gamma Stimulation" section. For example, taking into account age-related factors, switching from stroboscopic stimulation to stimulation with two or more alternating colors, taking into account the metabolic status of patients, etc.

Creating a table that takes into account absolutely all the side effects of gamma stimulation (even if it were possible) would exceed the scope of a single study.

  1. Authors make a great overview of the field with multiple details, now they need to write a section, which will summarize all the beneficial effects, protocols/waves needed for it, studies supporting it and what can be most adequate plan to use all known wavehterapies for the health, longevity and quality of life

Reply 5:

We thank the reviewer for this insightful comment. We dedicated an extensive portion of the review (more than a third of the entire text) to methods and examples of optimizing gamma stimulation and minimizing its side effects. Additionally, we added a chapter describing the main planned clinical trials of gamma stimulation devices in humans.

  1. Please reconcile wavelength claims by explaining when red/long wave vs white/500-600 nm stimuli are superior and why

Reply 6:

We agree with Reviewer, than the reduced efficacy of blue light for gamma stimulation compared to white light is consistent with classical concepts, however, traditionally, white and far-red colors are considered to be equally effective for γ stimulation. Our data indicate a greater efficacy of white light (sometimes containing near-red) compared to red. This phenomenon is easily explained by the higher sensitivity of the human eye to yellow-green light compared to red. Moreover, this sensitivity is expressed not only at the quantum level of the eye's receptor pigments, but also in the sensitivity of the central nervous system and its responses to color stimulation [73-75]. We added our exploration in the text.

MINOR

  1. Please remove ‘it is believed’ and change it to ‘it is currently suggested’ line 89

Reply 1: corrected

  1. Please standardize the numbering orders and sections to ensure correct cross-reference matches

Reply 2: corrected

Reviewer 3 Report

Comments and Suggestions for Authors

The manuscript contains a large amount of information and demonstrates considerable effort in data synthesis. Its major strength lies in the attempt to identify quantitative relationships governing γ-stimulation effectiveness. However, several issues should be addressed to improve clarity and methodological transparency

Comments

1) The manuscript repeatedly uses phrasing such as “we tested” or “we found,” implying that the authors conducted original experiments. This wording may confuse readers regarding data authorship. The authors should explicitly state that the work represents a quantitative synthesis or meta-analytical review based on previously published data.

2) The paper reports that 408 data points from 40 publications were analyzed, yet there is no description of:

  • how these studies were identified (databases, search terms, dates),
  • what inclusion/exclusion criteria were used,
  • whether duplicate or overlapping datasets were handled,
  • how different metrics were standardized before log transformation.

These details are essential for reproducibility

3) Sections 1–2 present an extensive general introduction to neural oscillations, gamma rhythms, and disease associations. While informative, this material overshadows the manuscript’s original contribution. The authors should condense the general background and place greater emphasis on the unique aspect of the study — namely, the quantitative comparison and optimization of γ-stimulation parameters.

4) In Figure 7 and related text, replace “γ-ray light stimulation” with “γ-frequency light stimulation,” as the former may be misleading.

Author Response

The manuscript contains a large amount of information and demonstrates considerable effort in data synthesis. Its major strength lies in the attempt to identify quantitative relationships governing γ-stimulation effectiveness. However, several issues should be addressed to improve clarity and methodological transparency

Comments

We thank Reviewer for carefully reading the manuscript and providing valuable recommendations. Responses to comments are highlighted in red.

1) The manuscript repeatedly uses phrasing such as “we tested” or “we found,” implying that the authors conducted original experiments. This wording may confuse readers regarding data authorship. The authors should explicitly state that the work represents a quantitative synthesis or meta-analytical review based on previously published data.

Reply 1:

We corrected indicated phrases, for example:

In the first stage of our analysis, we analyzed published data on the general effectiveness of the γ-stimulation method in various animal species.

We also add addition section, which described algorithm of literature searching and data processing.

According analyzes γ-stimulation was generally more effective in humans than in mice and marmosets.

Performed analyzes showed significant differences in the distribution patterns of γ-stimulation efficiency.

2) The paper reports that 408 data points from 40 publications were analyzed, yet there is no description of:

  • how these studies were identified (databases, search terms, dates),
  • what inclusion/exclusion criteria were used,
  • whether duplicate or overlapping datasets were handled,
  • how different metrics were standardized before log transformation.

These details are essential for reproducibility

Reply 2:

We thanks Reviewer for important comment and add more detained sections 3.1 and 3.2 about published data processing in the manuscript (see below):

3.1 Criteria for Including Publications in the Analysis

The work represents a quantitative synthesis or meta-analytical review based on previously published data. A literature search was performed using the publicly available databases PubMed (https://pubmed.ncbi.nlm.nih.gov/), Google Scholar (https://scholar.google.com/), and the websites of major publishers: Nature (https://www.nature.com/), MDPI (https://www.mdpi.com/), and others. The search was performed using the keywords 'gamma stimulation', 'gamma frequency stimulation', or '50 Hz brain rhythm stimulation'. To refine the search, we used combinations with additional keywords, such as: 'gamma stimulation' + 'disorders' (Alzheimer's disease, Parkinson's disease, etc.), 'gamma stimulation' + 'aging', 'gamma stimulation' + 'sleeping', 'gamma stimulation' + 'sound' or 'light', 'gamma stimulation' + 'human' or 'animals', etc. When including articles in the analysis, preference was given to articles published in the last 10 years (2015-2025) and 20 years (2005-2025). The shares of these works were 62% and 89%, respectively. Works published earlier (before 2005) were included if they contained pioneering data, important results for understanding the mechanisms of gamma rhythm functioning, or results that were not published in later periods. Publications that included completely identical experimental conditions to those already added, or those with insufficiently well-described experimental conditions (duration, frequency, nature of the stimulus, and its characteristics) were excluded from the analysis.

3.2. Data processing

As described previously, to enable comparison of disparate effects, we first converted to dimensionless quantities and calculated the effects of the treatments. We calculated the effects using formula (1)

(please, see attached file)

(1)

where A is the initial value of the measured characteristic (EEG potential, proportion of differentiated cells, mental performance scores, etc.), and ΔA is the value of the same parameter after γ-stimulation (in the same units as A). Expressing the values as percentages allows us to convert to dimensionless quantities and compare the effectiveness of γ-stimulation for different targets (Figure 10). Taking the absolute values allows us to adequately average the contributions of disparate effects. Further logarithmization allowed us to adequately assess the significance of statistical differences between groups, where effect sizes vary from units to hundreds of percent. The initial quantitative values of A and ΔA for the analysis were taken from the texts and/or graphs of the analyzed manuscripts. The calculated data were processed using non-parametric statistics. We used a Kruskal–Wallis ANOVA with a post-hoc Dunn's test to evaluate the significance of statistical differences. If the differences in median values between treatment groups were greater than would be expected by chance (p < 0.05), we proceeded with a pairwise multiple comparison using Dunn's test (all pairwise comparisons without control). P-values were provided for the results of the Dunn's test.

3) Sections 1–2 present an extensive general introduction to neural oscillations, gamma rhythms, and disease associations. While informative, this material overshadows the manuscript’s original contribution. The authors should condense the general background and place greater emphasis on the unique aspect of the study — namely, the quantitative comparison and optimization of γ-stimulation parameters.

Reply 3

We've shortened the introduction, leaving only the section devoted to the relevance of the study and its purpose.

We've combined detailed information on the role of the gamma rhythm in CNS functioning and the main mechanisms of its regulation into the section “2. General functions of γ-rhythms, mechanism of regulations”.

4) In Figure 7 and related text, replace “γ-ray light stimulation” with “γ-frequency light stimulation,” as the former may be misleading.

Reply 4

Corrected

Reviewer 4 Report

Comments and Suggestions for Authors

Thank you for the opportunity to review this work, which reviews gamma stimulation of the brain in multiple contexts. The paper is of interest, and nicely synthesizes work in the field to provide a good perspective on its current state.

I have some comments on the wording, descriptions, and minor comments on structure and content. 

Please find below specific comments.

Abstract:

The abstract in its current state is very blunt. Most phrases simply state results (e.g., 'Visual stimulation protocols using white light 40 are the most effective in humans'). Typically the reader would expect that the abstract introduces the topic, the methods used, and the key findings with some brief discussion. Rephrasing such as: 'Here, we found that experiments using white light were more effective than etc...' is more honest and clear regarding what is known, versus what is reviewed. This is an important difference as it changes the meaning of the phrases and tone of the information. 

Introduction:

1. Line 53: What does 'Non-invasive physical agricultural methods' mean in this context, and how does this relate to chemical and medicinal methods? This statement actually reads as if modern non-invasive approaches are replacing medicinal methods, which suggests that non-invasive approaches are not medicinal?

2. Line 74: Neural oscillations do not 'control' the synchronization of neural impulses, but rather are the product of it. 

3. Figure 1 is missing legends in the figure to indicate which line is which topic. I see this is in the legend, but I do not understand the reason for the text in the figure itself. The accession date for these data was also extracted in the future? (22 Nov 2025). The y axis should read 'Number of publications per year'.

4. Figures 2-5,10 use a Kurskal-wallis ANOVA and reports p value at the pair-wise level. Is the p value for the overall (group-level/methods-level) anova? is the pair-wise post-hoc test corrected for multiple comparisons? More clear methods are needed here. 

5. Figure 3 should read 'combined' not 'combine/conbine'. 

6. Figures 6 and 7 uses a comma for the decimal location, however the other figures use a period (P < 0.05), this should be standardized. The significance of these relationships should also be stated. 

7. Figure 8: this is an important figure. Could the authors also provide some indication of the significance of the effect module %? This could be done by indicating if z-scored effects exceeded significance thresholding at a certain age/wavelength point. On its own it is informative, but it is difficult to discern if the areas of larger effect are simply an artefact of scaling or are indeed significant. 

8. Likewise with Figure 9, there are interesting patterns here, however it is difficult to determine if the patterns are significant or simply a result of scaling. It would be an interesting and important addition to determine if for example 40 Hz at 0.25 years in Figure 9E is significantly more effective than other combinations. 

9. Line 731: 'It is activated when...' is not a proper phrase. I understand that this refers the the CEN, however this should be written more explicitely. Indeed section 4.3.2.1. is vague, short, and not very complementary to the rest of the manuscript. 

10. Line 995: What does the phrase mean ' optimal colour ... was differ' ?

Minor comments:

  1. line 167, 'In 2009' is missing the 'I'.

Comments on the Quality of English Language

Some general phrasing needs to be improved. In some sections there are no issues, however in the introduction and discussion there are several incomplete/odd sentences that are missing words, nouns, etc...

A thorough review of the language is needed to ensure completeness. 

Author Response

Thank you for the opportunity to review this work, which reviews gamma stimulation of the brain in multiple contexts. The paper is of interest, and nicely synthesizes work in the field to provide a good perspective on its current state.

I have some comments on the wording, descriptions, and minor comments on structure and content. 

Please find below specific comments.

Reply:

We thank Reviewer for carefully reviewing the manuscript and providing valuable recommendations. Responses to comments are highlighted in red.

Abstract:

The abstract in its current state is very blunt. Most phrases simply state results (e.g., 'Visual stimulation protocols using white light 40 are the most effective in humans'). Typically the reader would expect that the abstract introduces the topic, the methods used, and the key findings with some brief discussion. Rephrasing such as: 'Here, we found that experiments using white light were more effective than etc...' is more honest and clear regarding what is known, versus what is reviewed. This is an important difference as it changes the meaning of the phrases and tone of the information. 

Reply:

We corrected the abstract and change concrete results by more general conclusions.

The γ-rhythm plays a key role in coordinating the activity of the major brain systems and facilitating higher-level neurological processes. Several pathological conditions are associated with impaired generation or regulation of γ-oscillations. Modulating the γ-rhythm using periodic signals is considered a potential way to halt and/or treat such neurodegenerative processes. Despite the extensive knowledge gained in this field over the last 70 years, a unified theory linking the effectiveness of γ-stimulation to the characteristics of the stimulus and the stimulated remains elusive. In this review, we conducted a quantitative analysis of these relationships. The γ-stimulation effectiveness depends on species, age, frequency and stimulus type. Here, we found that experiments using white light were more effective that red and infrared. The range of effective central frequencies depends on age. We also showed, that AD patients and mouse models respond differently to γ-stimulation, so careful selection of study subjects is essential when assessing therapeutic potential. This review also provides an overview of the mechanisms of γ-stimulation and makes recommendations for optimising the method based on these mechanisms. Our findings may be useful to understanding of γ-stimulation mechanisms, planning future experiments for research groups and identifying potential therapeutic γ-stimulation regimens.

Introduction:

  1. Line 53: What does 'Non-invasive physical agricultural methods' mean in this context, and how does this relate to chemical and medicinal methods? This statement actually reads as if modern non-invasive approaches are replacing medicinal methods, which suggests that non-invasive approaches are not medicinal?

Reply 1:

The sentence was corrected “Non-invasive physical methods are becoming increasingly prevalent in a variety of sectors, including industry, agriculture, and medicine. These modern approaches are being incorporated into our lives and are gradually replacing traditional chemical methods.”

  1. Line 74: Neural oscillations do not 'control' the synchronization of neural impulses, but rather are the product of it.

Reply 2:

We've clarified the phrase "They control (or are a product of)" but kept the basic meaning.

Cell synchronization can be achieved by a variety of mechanisms: synchronous or sequential release of mediators/neurotransmitters (this is probably the mechanism the esteemed Reviewer had in mind), as well as through changes in the action potential, as in the case of electrical activity of the heart or brain. The role of synchronous electrical oscillations in coordinating the activity of different brain structures is generally accepted (10.3390/math11153307; 10.1016/s1388-2457(02)00292-4; 10.1073/pnas.0305375101). In addition, it has been described that the gamma rhythm is capable of demonstrating local synchronism in different brain structures (10.3389/fnint.2013.00058).

  1. Figure 1 is missing legends in the figure to indicate which line is which topic. I see this is in the legend, but I do not understand the reason for the text in the figure itself. The accession date for these data was also extracted in the future? (22 Nov 2025). The y axis should read 'Number of publications per year'.

Reply 3:

We added more detailed descriptions in the legend. Text on the figure we used to more clear reading. The accession data was corrected.

Figure 1. The dynamics of global publication activity on specific topics in gamma stimulation research. The search results for the following keywords are presented: 'gamma stimulation' (maroon line and fill), 'gamma stimulation' + 'disorders' (orange line and fill), 'gamma stimulation' + 'aging' (yellow line and fill), 'gamma stimulation' + 'Alzheimer's disease' (green line and fill) and 'gamma stimulation' + 'Parkinson's disease' (blue line and fill). The data were extracted from PubMed (https://pubmed.ncbi.nlm.nih.gov; accession date: 22 October 2025).

  1. Figures 2-5,10 use a Kurskal-wallis ANOVA and reports p value at the pair-wise level. Is the p value for the overall (group-level/methods-level) anova? is the pair-wise post-hoc test corrected for multiple comparisons? More clear methods are needed here. 

Reply 4:

We used a Kruskal–Wallis ANOVA with a post-hoc Dunn's test to evaluate the significance of statistical differences. If the differences in median values between treatment groups were greater than would be expected by chance (p < 0.05), we proceeded with a pairwise multiple comparison using Dunn's test (all pairwise comparisons without control). P-values were provided for the results of the Dunn's test. Figure captures were supplemented by “* — p < 0.05, Kurskal-wallis ANOVA with post-hoc Dunn’s test”.

  1. Figure 3 should read 'combined' not 'combine/conbine'. 

Reply 5: Corrected

Figure 3. A comparison of the efficacy of gamma-stimulation methods for different species. Data are presented as median values and percentiles at the 10th, 25th, 75th and 90th percentiles. Each point corresponds to an experimental value taken from the literature. Each article may contain at least one experimental point. A total of 393 experimental values from 38 published studies were analysed. * — p < 0.05, Kruskal–Wallis ANOVA with post-hoc Dunn’s test.

  1. Figures 6 and 7 uses a comma for the decimal location, however the other figures use a period (P < 0.05), this should be standardized. The significance of these relationships should also be stated.

Reply 6:

We thank Reviewer for carefully reading of the text and figures. Corrected

  1. Figure 8: this is an important figure. Could the authors also provide some indication of the significance of the effect module %? This could be done by indicating if z-scored effects exceeded significance thresholding at a certain age/wavelength point. On its own it is informative, but it is difficult to discern if the areas of larger effect are simply an artefact of scaling or are indeed significant. 

Reply 7:

We thank the Reviewer for their valuable recommendation.

To assess the significance of the data in the case of constructing 3D maps, we calculated z-scores according to the formula (2)

                                     (please, see attached file)                             (2)

when z is zi-score of individual i value, x is individual i value, μ is mean, σ is standard deviation. Statistically significant differences are those in which zi≥2.

According to our calculations, for mice, the log10 threshold for effects that differed significantly in z-score was 2.1. For humans, this value was slightly higher at 2.4. In both cases, the orange and red colours on the 3D maps correspond to areas with a z-score≥2 (p<0.05). Consequently, the main patterns we found are statistically significant.

This clarification has been added to the manuscript, and we will apply the z-score calculation approach in the future.

  1. Likewise with Figure 9, there are interesting patterns here, however it is difficult to determine if the patterns are significant or simply a result of scaling. It would be an interesting and important addition to determine if for example 40 Hz at 0.25 years in Figure 9E is significantly more effective than other combinations. 

Reply 8:

To assess the statistical significance of the observed effect dependences on age and wavelength, we calculated z-score values for each effect modules value (as described above). In this case, z-scores >2 were achieved for log10 effect sizes ≥1.95 (red) and ≥2.95 (yellow, orange, and red) for mice and humans, respectively. Therefore, the main relationships and patterns we found, shown in Figure 9, are statistically significant.

This clarification was added to the text of the article.

  1. Line 731: 'It is activated when...' is not a proper phrase. I understand that this refers the the CEN, however this should be written more explicitely. Indeed section 4.3.2.1. is vague, short, and not very complementary to the rest of the manuscript. 

Reply 9:

We corrected the phrase. “This neural network provides a solution of task that requires concentration, planning or problem-solving needs to be completed”.

Surprisingly, despite the significant importance of understanding that the brain functions as a network and that gamma oscillations mediate communication between different brain regions, we found no studies in the literature on the effects of gamma stimulation on the activity of known neural networks. The few literature sources we were able to find demonstrating the possibility of such an effect do not yet provide a comprehensive picture and are quite sparse. We would like to draw the attention of researchers to this extremely interesting area of research.

Here we present information on three of the most studied neural networks and our findings on their relationship with γ-stimulation.

Thus, it is clear that research is limited, and at present, only the possibility of such an effect exists. Nevertheless, we felt it was necessary to include this information in this article. Since it has been established that the functional activity of these neural networks is altered in a wide range of diseases, establishing the effect of gamma stimulation on their functioning could be very promising and open up additional possibilities for non-drug therapy.

  1. Line 995: What does the phrase mean ' optimal colour ... was differ' ?

Reply 10

The phrase was corrected to:

“The optimal colour, intensity and frequency of a flickering light stimulus for activating gamma waves differed between young adults (mean age 24 years) and more elderly persons.”

Minor comments:

  1. line 167, 'In 2009' is missing the 'I'.

Corrected

Comments on the Quality of English Language

Some general phrasing needs to be improved. In some sections there are no issues, however in the introduction and discussion there are several incomplete/odd sentences that are missing words, nouns, etc...

A thorough review of the language is needed to ensure completeness. 

Reply:

The text has been proofread and double-checked. Corrected words and sentences are highlighted in red.

Round 2

Reviewer 2 Report

Comments and Suggestions for Authors

I am thanking authors for careful evaluation of my comments and for implementing them in the manuscript.

I recommend accepting the manuscript in current form after finalising the translation typo  the reply 4 from 

Reviewer 3 Report

Comments and Suggestions for Authors

All reviewer comments were carefully addressed, and the manuscript has been significantly improved. I recommend acceptance in its current form.